# A theory of working memory without consciousness or sustained activity

**Darinka Trübutschek[1,2,3]\***, **Sébastien Marti[3]**, **Andrés Ojeda[4]**, **Jean-Rémi King[5,6]**, **Yuanyuan Mi[7]**, **Misha Tsodyks[8,9]**, **Stanislas Dehaene[3,10]**

[1]Ecole des Neurosciences de Paris Ile-de-France, 15 rue de l'Ecole de médecine, Paris, France; [2]Université Pierre et Marie Curie, 4 Place Jussieu, Paris, France; [3]Cognitive Neuroimaging Unit, CEA DSV/I2BM, INSERM, Université Paris-Sud, Université Paris-Saclay, NeuroSpin center, Gif/Yvette, France; [4]Department of Zoology, University of Oxford, Oxford, United Kingdom; [5]Department of Psychology, New York University, New York, United States; [6]Frankfurt Institute for Advanced Studies, Frankfurt, Germany; [7]Brain Science Center, Institute of Basic Medical Sciences, Beijing, China; [8]Department of Neurobiology, Weizmann Institute of Science, Rehovot, Israel; [9]Department of Neuroscience, Columbia University, New York, United States; [10]Collège de France, 11 Place Marcelin Berthelot, Paris, France

**\*For correspondence:**
darinkat87@gmail.com

**Competing interests:** The authors declare that no competing interests exist.

**Abstract** Working memory and conscious perception are thought to share similar brain mechanisms, yet recent reports of non-conscious working memory challenge this view. Combining visual masking with magnetoencephalography, we investigate the reality of non-conscious working memory and dissect its neural mechanisms. In a spatial delayed-response task, participants reported the location of a subjectively unseen target above chance-level after several seconds. Conscious perception and conscious working memory were characterized by similar signatures: a sustained desynchronization in the alpha/beta band over frontal cortex, and a decodable representation of target location in posterior sensors. During non-conscious working memory, such activity vanished. Our findings contradict models that identify working memory with sustained neural firing, but are compatible with recent proposals of 'activity-silent' working memory. We present a theoretical framework and simulations showing how slowly decaying synaptic changes allow cell assemblies to go dormant during the delay, yet be retrieved above chance-level after several seconds.

## Introduction

Prominent theories of working memory require information to be consciously maintained (*Baars and Franklin, 2003*; *Baddeley, 2003*; *Oberauer, 2002*). Conversely, influential models of visual awareness hold information maintenance as a key property of conscious perception, highlighting synchronous thalamocortical activity (*Tononi and Koch, 2008*), cortical recurrence (*Lamme and Roelfsema, 2000*), or the sustained recruitment of parietal and dorsolateral prefrontal regions (i.e., the same areas as in working memory; *Naghavi and Nyberg, 2005*) in a global neuronal workspace (*Dehaene and Changeux, 2011*, *2001*). Experimentally, non-conscious priming only lasts a few hundred milliseconds (*Dupoux et al., 2008*; *Greenwald et al., 1996*) and unseen stimuli typically fail to induce late and sustained cerebral responses (*Dehaene et al., 2014*). Conscious perception, in contrast, exerts a durable influence on behavior, accompanied by sustained neural activity (*King et al., 2014*; *Salti et al., 2015*; *Schurger et al., 2015*). The hypothesis of an intimate coupling between

**eLife digest** Many everyday activities require you to store information in your brain for immediate use. For example, imagine that you are cooking a meal: You have to remember the ingredients, add them in the correct order, and operate the stove. This ability is called working memory.

Researchers have long believed that, whenever we store information in our working memory, we are conscious of that information. That is, if someone asks you, you can report the information. Scientists usually also think that working memory comes with constant brain activity. This means that for as long as you have to remember something, the cells in your brain that code for that information will be active.

Trübutschek et al. now show that we can sometimes store information in working memory without being conscious of it and without the need for constant brain activity. As part of the experiment, a barely visible square-shaped target was briefly flashed in 1 of 20 different locations on a computer screen. Human volunteers had to locate the square and indicate whether they had seen it or not. Importantly, they had to guess the location of the target whenever they had not seen it. While the volunteers performed this task, their brain activity was monitored using magnetoencephalography, a noninvasive technique that captures the magnetic fields created by electrical signals in the brain.

Even when the volunteers had not seen the target, they could often correctly guess where it had been up to four seconds later, more often than would be predicted by chance alone. The experiment ruled out the possibility that this so-called "blindsight" was simply due to the volunteers accidentally reporting not having seen a target, when they had actually seen it. It also excluded the possibility that the volunteers guessed the location long before they had to report it and simply consciously stored that guess. Instead, without the participant knowing, the brain appears to have stored the target location in working memory using parts of the brain near the back of the head that process visual information. Importantly, this non-conscious storage did not come with constant brain activity, but seemed to rely on other, "activity-silent" mechanisms that are hidden to standard recording techniques.

Although Trübutschek et al. show that the brain can unknowingly store information, they did not test other aspects of working memory. Future studies are needed to examine whether the brain can also non-consciously manipulate or use information in its working memory. In addition, future research also needs to investigate the exact mechanism that stores information without constant brain activity.

conscious perception and working memory is thus grounded in theory and supported by numerous empirical findings.

Recent behavioral and neuroimaging evidence, however, has questioned this prevailing view by suggesting that working memory may also operate non-consciously. Unseen stimuli may influence behavior for several seconds (*Bergström and Eriksson, 2015*; *Soto and Silvanto, 2014*). *Soto et al. (2011)*, for instance, showed that participants recalled the orientation of a subjectively unseen Gabor cue above chance-level after a 5s-delay. Functional magnetic resonance imaging suggests that pre-frontal activity may underlie such non-conscious working memory (*Bergström and Eriksson, 2014*; *Dutta et al., 2014*).

The verdict for non-conscious working memory is far from definitive, however. Delayed performance with subjectively unseen stimuli was barely above chance (*Soto et al., 2011*) and could have arisen from a small percentage of errors in visibility reports, with subjects miscategorizing a seen target as unseen (miscategorization hypothesis). If this were the case, then the blindsight trials, on which subjects correctly identified the target while denying any subjective awareness of the stimulus, should display similar, if not identical, neural signatures and contents as the seen trials. Alternatively, participants could also have ventured a guess about the target as soon as it appeared and consciously maintained this early guess (conscious maintenance hypothesis). Many priming studies have shown that fast guessing results in above-chance objective performance with subjectively unseen

stimuli (*Merikle et al., 2001*). The observed blindsight effect would then reflect a normal form of conscious working memory (*Stein et al., 2016*). This alternative hypothesis is hard to eliminate on purely behavioral grounds; it can only be rejected by tracking the dynamics of working memory activity, for instance using brain-imaging, and determining whether this activity occurs immediately after the target even on unseen trials.

Here, we set out to address these issues, focusing on four main objectives: First, we probed the replicability of the long-lasting blindsight effect reported by *Soto et al. (2011)* as well as its robustness with respect to interference from distraction and a conscious working memory load in order to delineate it from other forms of prolonged iconic or sensory memory. Second, we interrogated the link between conscious perception and conscious working memory, examining whether the maintenance period in working memory could be likened to a prolongation of a conscious episode. Third, we tested the reality of non-conscious working memory by systematically examining the neural correlates of the blindsight effect and using them to assess the above two alternative hypotheses (the miscategorization and conscious maintenance hypothesis). Lastly, we propose a neuronal theory to offer a mechanistic account of conscious and non-conscious working memory.

## Results

We combined magnetoencephalography (MEG) with a spatial masking paradigm to assess working memory performance under varying levels of subjective visibility (*Figure 1A* and Materials and methods). On 80% of the trials, a target square was flashed in 1 of 20 locations and then masked. Subjects were asked to localize the target after a variable delay (2.5–4.0 s) and to rate its visibility on a scale from *1* (not seen) to *4* (clearly seen). On the remaining 20% of trials, the target was omitted, allowing us to contrast brain activity between target-present and -absent trials. A visible distractor square was presented 1.5 s into the delay period on half the trials, challenging participants' resistance to distraction and enabling us to evaluate the robustness of the blindsight effect behaviorally. In addition to this working memory task, subjects also completed a perception-only control condition without the delay and target-localization periods (perception task), so that we could isolate brain activity specific to conscious perception (without a working memory requirement) and investigate its link with working memory.

### Behavioral maintenance and shielding against distraction

We first examined objective performance in the working memory task as a function of target visibility. Overall, subjects reported the exact target location with high accuracy on seen trials (collapsed across visibility ratings > *1*: $M_{correct}$ = 69.1%, $SD_{correct}$ = 17.4%; chance = 5%; t(16) = 15.2, p<0.001, 95% CI = [55.2%, 73.1%]; Cohen's d = 3.7). As subjective visibility of the target increased from glimpsed (visibility = *2*) to clearly seen (visibility = *4*), there was a corresponding monotonic increase in accuracy (*Figure 1B*; ps<0.05 for all pair-wise comparisons). Crucially, performance remained above chance even on unseen trials (rating = *1*: $M_{correct}$ = 22.4%, $SD_{correct}$ = 13.8%; t(16) = 5.2, p<0.001, 95% CI = [10.3%, 24.4%]; Cohen's d = 1.3). This blindsight remained substantial after a 4s-delay ($M_{correct}$ = 21.1%, $SD_{correct}$ = 14.7%; t(16) = 4.5, p<0.001, 95% CI = [8.5%, 23.7%]; Cohen's d = 1.0).

Spatial distributions of participants' responses were concentrated around the target (*Figure 2A*). To correct for small errors in localization, we computed the rate of correct responding with a tolerance of two positions (±36°) surrounding the target location. In subjects displaying above-chance blindsight (chance = 25%; p<0.05 in a $\chi^2$-test; n = 13), we estimated the precision of working memory as the standard deviation of the distribution within this tolerance interval (Materials and methods). Performance was better on seen than on unseen trials, both in terms of rate of correct responding (F(1, 16) = 198.5, p<0.001; partial $\eta^2$ = 0.925) and precision (F(1, 12) = 36.7, p<0.001; partial $\eta^2$ = 0.754). There was neither an effect of the distractor on these measures (all ps>0.079), nor any significant interactions between distractor and visibility (all ps>0.251), indicating that distractor presence did not affect retention for seen or unseen targets. Restricting the analyses to trials within one position of the actual target location (±18°) or to the subgroup of 13 subjects included in the MEG analyses did not change these findings qualitatively.

While target detection d' exceeded chance-level (M = 1.5, SD = 0.7; t(16) = 8.9, p<0.001, 95% CI = [1.2, 1.9]; Cohen's d = 2.1) and correlated with accuracy and the rate of correct responding on

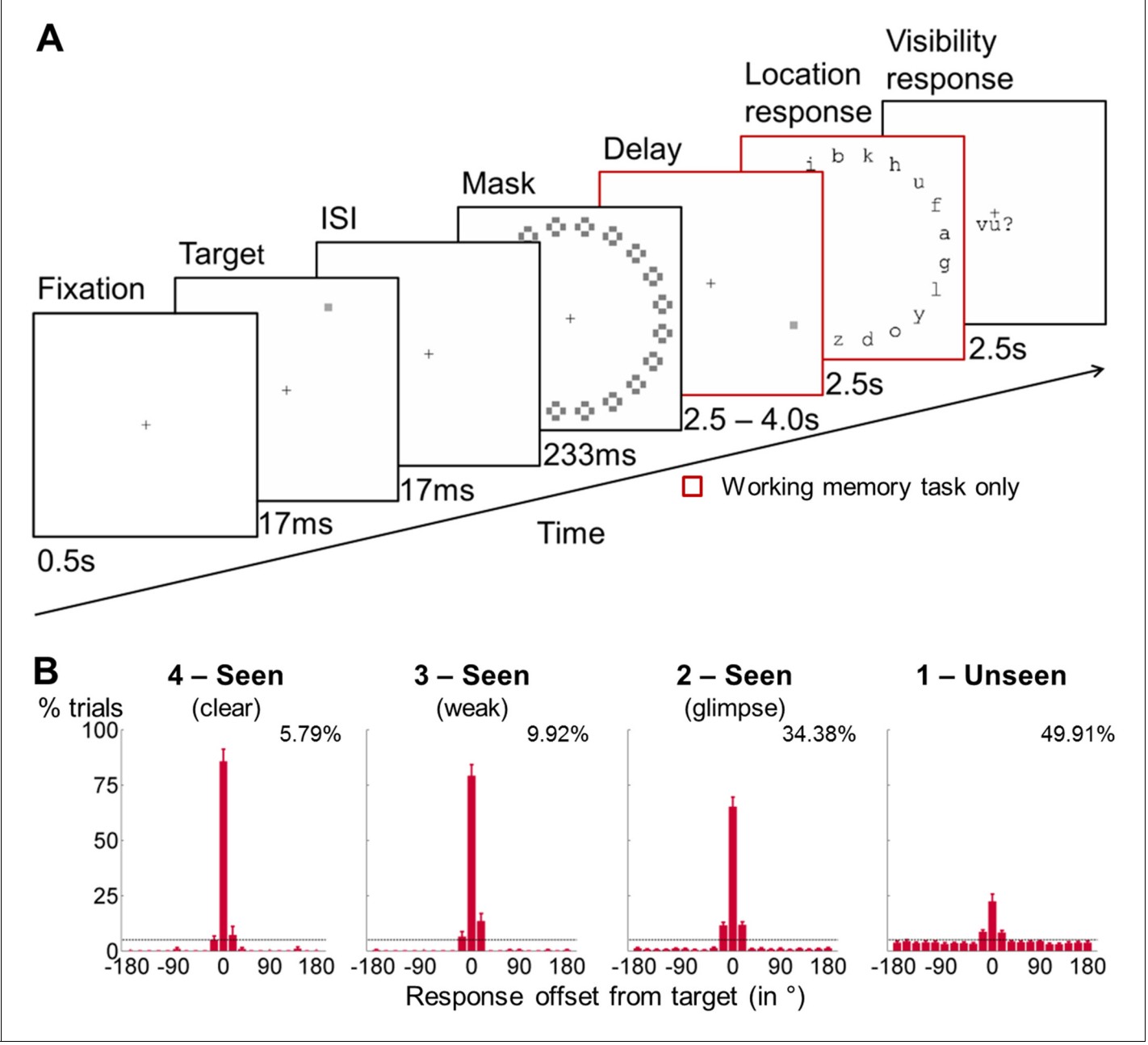

**Figure 1.** General experimental design and behavioral performance in the working memory task. (**A**) Experimental design. A subsequently masked target square was flashed in 1 out of 20 positions. Subjects were asked to report this location after a delay of up to 4 s and to rate the visibility of the target on a 4-point scale. A visible distractor square with features otherwise identical to the target was shown on 50% of the trials during the retention period (at 1.75 s). In a perception-only control condition, the maintenance phase and location response were omitted, and subjects assessed the visibility of the target immediately after the mask. (**B**) Spatial distributions of forced-choice localization performance in the working memory task (experiment 1; 0 = correct target location; positive = clockwise offset). Error bars indicate standard error of the mean (SEM) across subjects. The horizontal, dotted line illustrates chance-level at 5%. Percentages show proportion of target-present trials from a given visibility category. Due to low number of trials in individual visibility ratings *2*, *3*, and *4*, all *seen* categories were collapsed for analyses.

seen trials (both Pearson rs > 0.762, both ps<0.001), there was no relationship between our participants' sensitivity to the target and any of our performance measures on the unseen trials (all Pearson rs < 0.342, all ps>0.179; *Figure 2—figure supplement 1A*). Thus, target visibility predicted performance in the objective working memory task only on seen trials, but not on unseen trials.

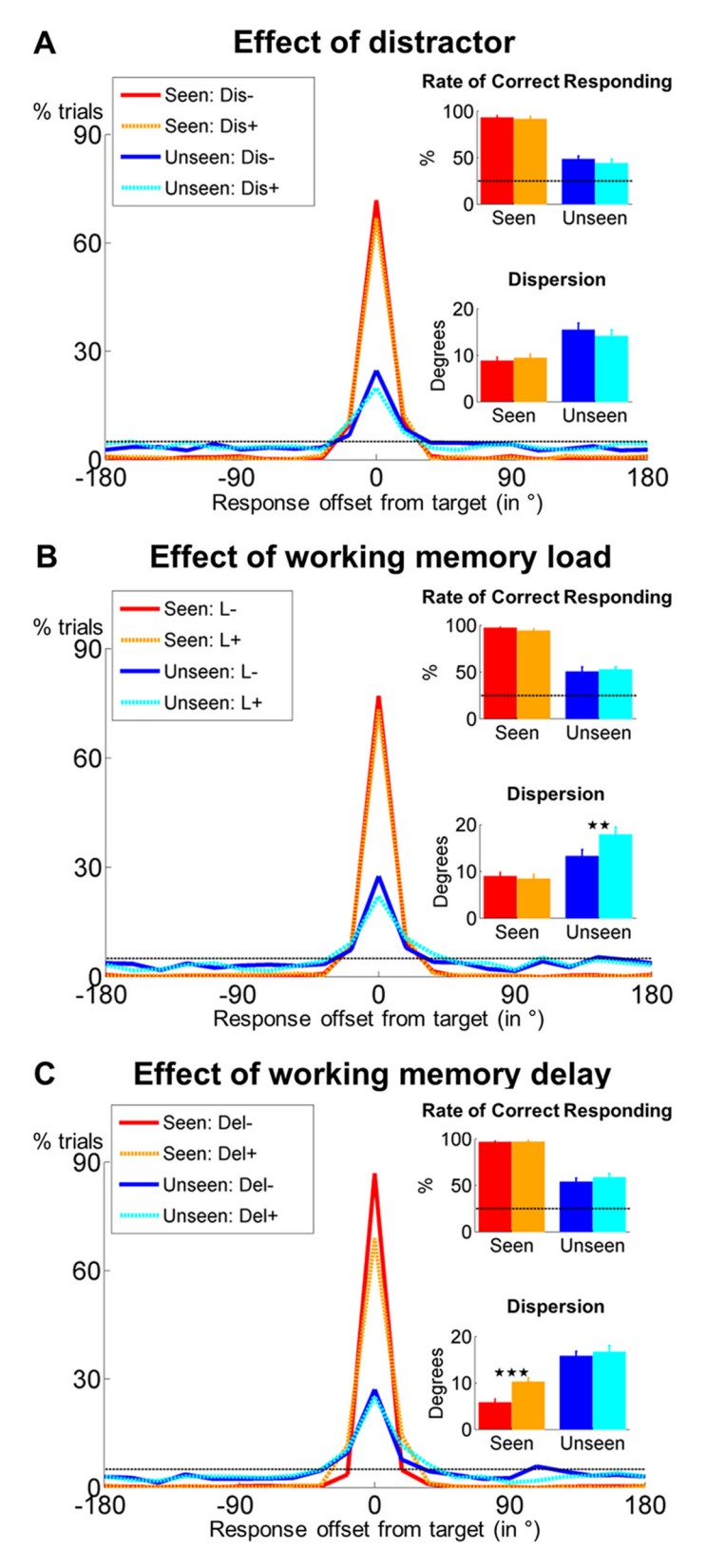

**Figure 2.** Behavioral evidence for non-conscious working memory. Spatial distributions of responses (0 = correct target location; positive = clockwise offset) as a function of visibility and distractor presence (**A**), conscious working memory load (**B**) and delay duration (**C**). Insets show rate of correct
*Figure 2 continued on next page*

*Figure 2 continued*

responding (within ±2 positions of actual location) and precision of working memory representation separately for seen and unseen trials. Error bars represent standard error of the mean (SEM) across subjects and horizontal, dotted line indicates chance-level (5%). *p<0.05, **p<0.01, and ***p<0.001 in a paired sample *t*-test. Del = delay, Dis = distractor, L = load.

The following figure supplement is available for figure 2:

**Figure supplement 1.** Perceptual sensitivity does not correlate with working memory performance on unseen trials.

Overall, these results confirm, with much higher non-conscious performance, the observations of previous studies (*Soto et al., 2011*): Non-conscious information may be maintained for up to 4 s and successfully shielded against distraction from a salient visual stimulus, independently of overall subjective visibility.

## Resistance to conscious working memory load and delay duration

To probe the similarity between conscious working memory and the observed long-lasting blindsight effect, in a second behavioral experiment with 21 subjects, we examined whether imposing a load on conscious working memory (remembering digits) affected non-conscious performance. On each trial, 1 (low load) or 5 (high load) digits were simultaneously shown for 1.5 s, followed by a 1s-fixation period and the same sequence of events (target and mask) as in experiment 1. After a variable delay (0 or 4 s), participants had to (1) localize the target, (2) recall the digits in the correct order, and (3) rate target visibility.

Subjects again chose the exact target position with high accuracy on seen trials ($M_{correct}$ = 77.8%, $SD_{correct}$ = 13.9%) and remained above chance on unseen trials ($M_{correct}$ = 25.6%, $SD_{correct}$ = 11.8%; chance = 5%; $t(18)$ = 7.6, $p<0.001$, 95% CI = [14.9%, 26.3%]; Cohen's d = 1.7). While, as in experiment 1, cue detection d' was greater than chance (M = 1.7, SD = 0.8; $t(20)$ = 10.2, $p<0.001$, 95% CI = [1.4, 2.1]; Cohen's d = 2.2), no correlations were observed with objective task performance on the unseen trials (all Pearson rs < 0.366, all ps>0.115; seen trials: all Pearson rs > 0.443, all ps<0.051; *Figure 2—figure supplement 1B*). As expected, participants were better at recalling 1 rather than 5 digits in the correct order (M = 93.3% vs. 89.5%, $F(1, 17)$ = 4.7, $p=0.045$), irrespective of target visibility or delay duration (all ps>0.135).

Analyzing only the trials with correctly recalled digits, we observed an impact of load on the precision with which target location was retained ($F(1, 13)$ = 7.3, $p=0.018$; partial $\eta^2$ = .360). Crucially, load modulated the relationship between precision and visibility (interaction $F(1, 13)$ = 8.7, $p=0.011$; partial $\eta^2$ = .400), with no effect on seen ($t(13)$ = 0.6, $p=0.561$) and a strong reduction of precision on unseen trials ($t(13)$ = −3.6, $p=0.004$). There was no effect of working memory load on the rate of correct responding (all ps>0.229; *Figure 2B*).

Delay duration (0 or 4 s) also did not influence the rate of correct responding (all ps>0.082; *Figure 2C*). It did, however, affect overall precision ($F(1, 15)$ = 9.3, $p=0.008$; partial $\eta^2$ = .383) and the relationship between precision and visibility (interaction $F(1, 15)$ = 5.2, $p=0.037$; partial $\eta^2$ = .259). This interaction was driven by higher precision on no-delay than on 4s-delay trials, exclusively when subjects had seen the target ($t(15)$ = −5.7, $p<0.001$; unseen trials: $t(15)$ = −0.6, $p=0.559$).

Overall, these results highlight the replicability and robustness of the long-lasting blindsight effect and suggest that it does not just constitute a prolonged version of iconic memory: Even in the presence of a concurrent conscious working memory load, unseen stimuli could be maintained, with no detectable decay as a function of delay. However, the systems involved in the short-term maintenance of conscious and non-conscious stimuli interacted, because a conscious verbal working memory load diminished the precision with which non-conscious spatial information was maintained.

## Similarity of conscious perception and conscious working memory

To tackle our second objective – a detailed examination of the link between conscious perception and conscious working memory –, we turned to our MEG data and first ensured that the mechanisms underlying conscious perception were stable across experimental conditions. The subtraction of the

event-related fields (ERFs) evoked by unseen trials from those evoked by seen trials revealed similar topographies for the perception and working memory task (*Figure 3A*): Starting at ~300 ms and extending until ~500 ms after target onset, a response emerged over right parieto-temporal magnetometers. This divergence resulted primarily from a sudden increase in activity on seen trials ('ignition') in the perception ($p_{FDR}$<0.05 from 384 to 416 ms and from 504 to 516 ms) and working memory task ($p_{FDR}$<0.05 from 328 to 364 ms and from 396 to 404 ms; *Figure 3B*). The observed topographies and time courses fall within the time window of typical neural markers of conscious perception, including the P3b (e.g., *Del Cul et al., 2007*; *Salti et al., 2015*; *Sergent et al., 2005*). Consciously perceiving the target stimulus therefore involved comparable neural mechanisms, irrespective of task.

We next directly probed the relationship between conscious perception and information maintenance in conscious working memory. Does the latter reflect a prolonged conscious episode, or does it involve a distinct set of processes recruited only during the retention phase? If conscious working memory can indeed be likened to conscious perception, one might expect the same patterns that index such perception to be sustained throughout the working memory maintenance period. Linear multivariate pattern classifiers were trained to predict visibility (seen or unseen) from MEG signals separately for each task. Classification performance was assessed during an early time period (100–300 ms), the critical P3b time window (300–600 ms), and the first (0.6–1.55 s) and second part (1.55–2.5 s) of the delay period.

Decoding of the visibility effect was comparable in the two tasks (*Figure 3C* and *Supplementary file 1*): Classification performance rose sharply between 100 and 300 ms and peaked during the P3b time window (all ps<0.007, except 100–300 ms in the working memory task, where p=0.066). It then decayed slowly from ~1 s onwards in both tasks, yet remained above chance during the 0.6–1.55 s interval (all ps<0.001). Similar time courses were also observed when training in one task and testing for generalization to the other. Though rapidly dropping to chance-level after ~1 s, classifiers trained in the perception task performed above chance during the first three time windows on working memory trials (and vice versa; all ps<0.014), indicating that, early on, both tasks recruited similar brain mechanisms.

Temporal generalization analyses (*King and Dehaene, 2014*) were used to evaluate the onset and duration of patterns of brain activity. If working memory were just a prolonged conscious episode, classifiers trained at time points relevant to conscious perception (e.g., the P3b window) should generalize extensively, potentially spanning the entire delay. Our findings supported this hypothesis only in part. The temporal generalization matrix for the working memory task presented as a thick diagonal, suggesting that brain activity was mainly characterized by changing, but long-lasting patterns. Though failing to achieve statistical significance over the entire 0.6–1.55 s interval (all ps>0.101), at a more lenient, uncorrected threshold, classifiers trained during the P3b time window (300–600 ms) in the working memory task remained weakly efficient until ~692 ms (AUC = 0.54 ±- 0.02, $p_{uncorrected}$=0.023). Similarly, classifiers trained during the same time period in the perception task and tested in the working memory task persisted up to ~860 ms (AUC = 0.53 ± 0.01, $p_{uncorrected}$=0.028). Brain processes deployed for the conscious representation of the target were thus partially sustained during the working memory delay. The reverse analysis, in which we trained classifiers during the retention period in the working memory task (0.8–2.5 s), did not reveal any generalization to the P3b time window in the perception task (p= 0.101).

These results confirm that seeing the target entailed a similar unfolding of neural events in two task contexts: Conscious perception primarily consisted in a dynamic series of partially overlapping information-processing stages, each characterized by temporary, metastable patterns of neural activity. The same neural codes appeared to be recruited at the beginning of the maintenance period (up to ~1 s). As such, these findings corroborate previous accounts linking conscious perception to an 'ignition' of brain activity (*Del Cul et al., 2007*; *Gaillard et al., 2009*; *Salti et al., 2015*; *Sergent et al., 2005*) and suggest that, in part, working memory implies the prolongation of a conscious episode, and, in part, a succession of additional processing steps.

## A sustained decrease in alpha/beta power distinguishes conscious working memory

Our focus so far has been on evoked brain activity. However, other reliable neural signatures of conscious perception have been identified in the frequency domain (*Gaillard et al., 2009*; *Gross et al.,*

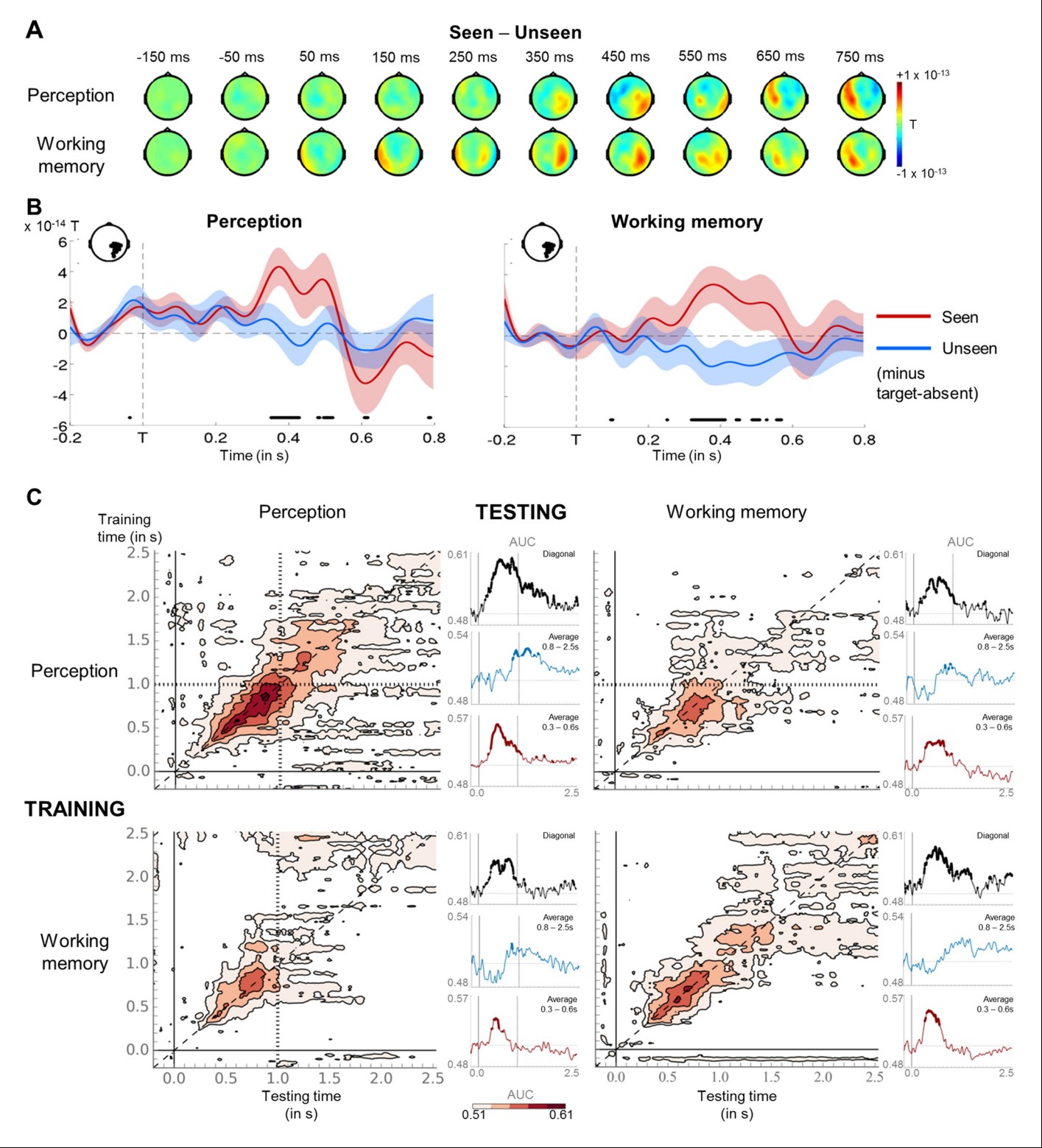

**Figure 3.** Neural signatures for conscious perception and maintenance in working memory. (**A**) Sequence of brain activations (−200–800 ms) evoked by consciously perceiving the target in the perception (top) and working memory (bottom) task. Each topography depicts the difference in amplitude between seen and unseen trials over a 100 ms time window centered on the time points shown (magnetometers only). (**B**) Average time courses of seen and unseen trials (−200–800 ms) after subtraction of target-absent trials in a group of parietal magnetometers in the perception (left) and working memory (right) task. Shaded area illustrates standard error of the mean (SEM) across subjects. Significant differences between conditions are depicted

*Figure 3 continued on next page*

*Figure 3 continued*

with a horizontal, black line (Wilcoxon signed-rank test across subjects, uncorrected). For display purposes, data were lowpass-filtered at 8 Hz. T = target onset. (C) Temporal generalization matrices for decoding of visibility category as a function of training and testing task. In each panel, a classifier was trained at every time sample (y-axis) and tested on all other time points (x-axis). The diagonal gray line demarks classifiers trained and tested on the same time sample. Please note the event markers in any panel involving the perception task: Mean reaction time (target-present trials) for the visibility response is indicated as vertical and/or horizontal, dotted lines. Any classifier beyond this point only reflects post-visibility processes. Time courses of diagonal decoding and of classifiers averaged over the P3b time window (300–600 ms) and over the working memory maintenance period (0.8–2.5 s) are shown as black, red, and blue insets. Thick lines indicate significant, above-chance decoding of visibility (Wilcoxon signed-rank test across subjects, uncorrected, two-tailed except for diagonal). For display purposes, data were smoothed using a moving average with a window of eight samples. AUC = area under the curve.

*2007*; *King et al., 2016*; *Wyart and Tallon-Baudry, 2009*). We thus turned to time-frequency analyses and first contrasted seen trials with both our target-absent control condition as well as unseen trials in both tasks (*Figure 4A* and *Figure 4—figure supplement 1A*). In order to qualify as a signature of conscious perception, any candidate characteristic should exist in the perception-only control condition (without any working memory requirement) and be specific to seen trials. Cluster-based permutation analyses singled out a desynchronization in the alpha band (8–12 Hz) as the principal correlate of conscious perception in the perception task (seen – target-absent: $p_{clust}$=0.004; seen – unseen: $p_{clust}$=0.009), with seen trials displaying a strong decrease in power (relative to baseline) compared to either the target-absent or the unseen trials. Initially left-lateralized in centro-temporal sensors, this effect moved to fronto-central channels and extended between ~300 and 1700 ms. A similar, albeit later (500–1700 ms) and more bilateral fronto-central, desynchronization was also observed in the beta band (13–30 Hz; seen – target-absent: $p_{clust}$<0.001; seen – unseen: $p_{clust}$=0.01). No differences between the unseen and target-absent trials were found in the alpha ($p_{clust}$>0.676) or beta band ($p_{clust}$>0.226, apart from a short-lived, weak difference between ~0.9 and 1.3 s, where $p_{clust}$=0.020), suggesting that unseen trials strongly resembled trials without a target.

Most importantly, when comparing seen and target-absent/unseen trials in the working memory task, we again observed a similar, but now temporally sustained, pattern of alpha/beta band desynchronization (*Figure 4B* and *Figure 4—figure supplement 1B*). Starting at ~300 to 500 ms, seen targets evoked a power decrease in central, temporal/parietal, and frontal regions in the alpha (seen – target-absent: $p_{clust}$=0.003; seen – unseen: $p_{clust}$=0.003) and beta band (seen – target-absent: $p_{clust}$=0.009; seen – unseen: $p_{clust}$<0.001). Crucially, this desynchronization spanned the entire delay period and was specific to seen trials (*Figure 4A*), with no differences in power between the unseen and target-absent trials in either band (alpha: $p_{clust}$>0.729; beta: $p_{clust}$>0.657) and only a couple of interspersed periods of residual desynchronization persisting in the target-absent control trials. No task- or visibility-related modulations in power spectra were found in occipital areas, and the desynchronization originated primarily from a parietal network of brain sources (*Figure 4A and B*). In conjunction with the afore-mentioned results, these findings imply that alpha/beta desynchronization is a correlate of conscious perception (*Gaillard et al., 2009*) and a neural state common to conscious perception and conscious working memory.

## A distinct neurophysiological mechanism for non-conscious working memory

Having identified markers of conscious perception and working memory in both multivariate and time-frequency analyses, we can now test the reality of non-conscious working memory by confronting it with several alternative hypotheses. The miscategorization hypothesis suggests that the long-lasting blindsight resulted from a small set of seen trials erroneously labeled as unseen. Unseen correct trials should thus display similar neural signatures as seen trials, including a shared discriminative decoding axis and a desynchronization in the alpha/beta band. An analogous reasoning holds for the conscious maintenance hypothesis, according to which the observed blindsight effect arises from the conscious maintenance of an early guess: Conscious processing would occur on unseen trials and we should thus find a sustained decrease in alpha/beta power similar to the one on seen trials. Conversely, a clear distinction between brain responses on seen trials and on unseen (correct) trials

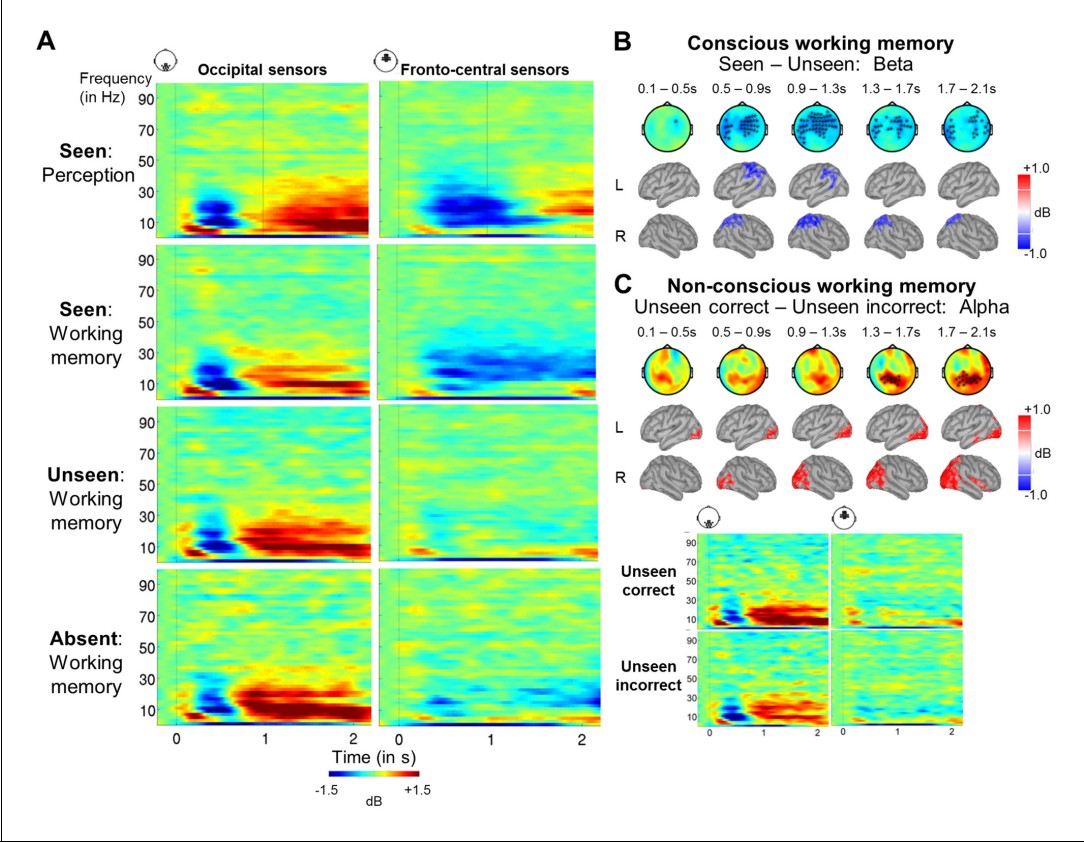

**Figure 4.** A sustained decrease in alpha/beta power as a marker of conscious working memory. (A) Average time-frequency power relative to baseline (dB) as a function of task and visibility category in a group of occipital (left) and fronto-central (right) magnetometers. Mean reaction time (target-present trials) for the visibility response in the perception task is indicated as a vertical, dotted line. (B) Beta band activity (13–30 Hz; 0–2.1 s) related to conscious working memory (seen – unseen trials) as shown in magnetometers (top) and source space (bottom; in dB relative to baseline). Black asterisks indicate sensors showing a significant difference as assessed by a Monte-Carlo permutation test. (C) Same as in (A) and (B) but for unseen correct and unseen incorrect trials in the alpha band (8–12 Hz).

The following figure supplements are available for figure 4:

**Figure supplement 1.** Alpha- and beta-band desynchronizations serve as a general signature of conscious processing and conscious working memory.

**Figure supplement 2.** Seen and unseen correct trials do not share the same discriminative decoding axis.

**Figure supplement 3.** Bayesian statistics for the time-frequency analyses.

would suggest that blindsight resulted from a distinct non-conscious mechanism of information maintenance.

We first probed the alternative hypotheses with the ERF data. Training a decoder to distinguish seen from unseen trials in the perception task and applying it to the unseen correct and incorrect trials in the working memory task, we directly assessed the classifier's ability to generalize from seen to unseen correct trials (accuracy decoder). If, indeed, the latter had actually been seen, such a decoder should look similar to the above-described generalization analysis, in which a classifier had been trained on seen/unseen trials in the perception task and tested on the same labels in the working memory task (visibility decoder). As shown in *Figure 4—figure supplement 2A*, this was not the case. Whereas the temporal generalization matrix for the visibility decoder presented as a thick diagonal, no discernable pattern emerged for the accuracy decoder. The time courses of diagonal decoding were also quite dissimilar. For the visibility decoder (see also above), classification

performance first rose above chance at ~148 ms (AUC = 0.54 ± 0.01, $p_{FDR}$=0.023), peaked at ~640 ms (AUC = 0.58 ± 0.02, $p_{FDR}$=0.001), and then decayed rapidly by ~1 s (first three time windows, all ps<0.001). In contrast, classification for the accuracy decoder was erratic and transient: It first sharply peaked at ~180 ms (AUC = 0.55 ± 0.01, $p_{uncorrected}$=0.037), dropped to chance-level, and then exceeded chance between ~372 and 724 ms with a peak at 444 ms (AUC = 0.57 ± 0.02, $p_{uncorrected}$=0.007). Much unlike any of the previous decoders involving the perception task, long after the visibility response, it rose a third time between ~1.44 and 1.74 s, peaking with similar magnitude as before at ~1.58 s (AUC = 0.57 ± 0.02, $p_{uncorrected}$=0.010; P3b and last time window: all ps<0.023). Although the level of noise evident in the accuracy decoder thus precludes any definitive conclusion, the visibility and accuracy decoders had little in common, rendering it unlikely for the unseen correct trials to have simply been mislabeled.

We next returned to our time-frequency analysis. When averaging over all unseen trials in the working memory task, there was no indication of a desynchronization remotely comparable to the one on seen trials (*Figure 4A* and *Figure 4—figure supplement 1C*). Indeed, Bayesian statistics indicated that, on the unseen trials, evidence for the null hypothesis (i.e., no relative change in alpha/beta power) was at least similar (at the very end of the epoch) or stronger than evidence for the alternative hypothesis. By contrast, on seen trials, evidence for the alternative hypothesis was always strongly favored (*Figure 4—figure supplement 3*). Even when analyzing the unseen correct trials separately, there was no appreciable trace of any alpha/beta desynchronization (*Figure 4C* and *Figure 4—figure supplement 3*). Only one short-lived effect, reversed relative to conscious trials, was observed in the alpha band ($p_{clust}$=0.040) in a set of posterior central sensors, corresponding to primarily occipital sources: Starting at ~1.5 s and extending until ~1.9 s, unseen correct trials exhibited a stronger *increase* in alpha power than their incorrect counterparts. Given the difference in performance on these two types of unseen trials, such small variations are not surprising and could, perhaps, reflect a stronger suppression of interference from the distractor on the unseen correct trials. Unseen correct trials thus appeared to be nearly indistinguishable from the unseen incorrect and target-absent trials.

As multivariate analyses might be more sensitive than univariate ones in detecting similarities between conditions, we also performed the above decoding analysis separately for average alpha (8–12 Hz) and beta (13–30 Hz) power. Overall, these analyses confirmed our previous findings, albeit more clearly so in the alpha than in the beta band. A visibility decoder trained on alpha power to distinguish seen from unseen trials in the perception task and tested in the working memory task again exhibited a thick diagonal, with above-chance decoding between ~180 ms and 1.18 s (first three time windows: all ps<0.016). There was no evidence for any generalization to the unseen correct trials (*Figure 4—figure supplement 2B*; all time windows: ps>0.211). Similarly, a visibility decoder trained on average beta power entirely failed to generalize to the unseen correct trials (*Figure 4—figure supplement 2C*; all time windows: ps>0.191). Considering the weak, although statistically significant (all four time windows, ps<=0.05), initial generalization from the perception to the working memory task, probably due to the slightly later onset of the beta desynchronization in the former, this failure is less informative than the one observed in the alpha band and should be replicated in future investigations.

Taken together, we found a clear distinction in the brain responses of seen and unseen (correct) trials. Converging evidence from our decoding analyses in the ERFs and alpha/beta band suggests that there was no apparent discriminative axis shared between the seen and the unseen correct trials. Similarly, the desynchronization in alpha/beta power characterizing the seen targets did not emerge on the unseen (correct) trials. These findings therefore argue against the miscategorization and conscious maintenance hypotheses and instead suggest that non-conscious working memory is a genuine phenomenon, distinct from conscious working memory.

## Contents of conscious and non-conscious working memory can be tracked transiently

We next set out to identify the neural mechanisms supporting both conscious and non-conscious working memory and first determined where and how the specific contents of working memory were stored. Circular-linear correlations between the amplitude of the ERFs and target location (across all working memory trials) revealed a strong and focal association (relative to a permuted null distribution) over posterior channels, starting at ~120 ms and lasting until 904 ms (early and P3b time

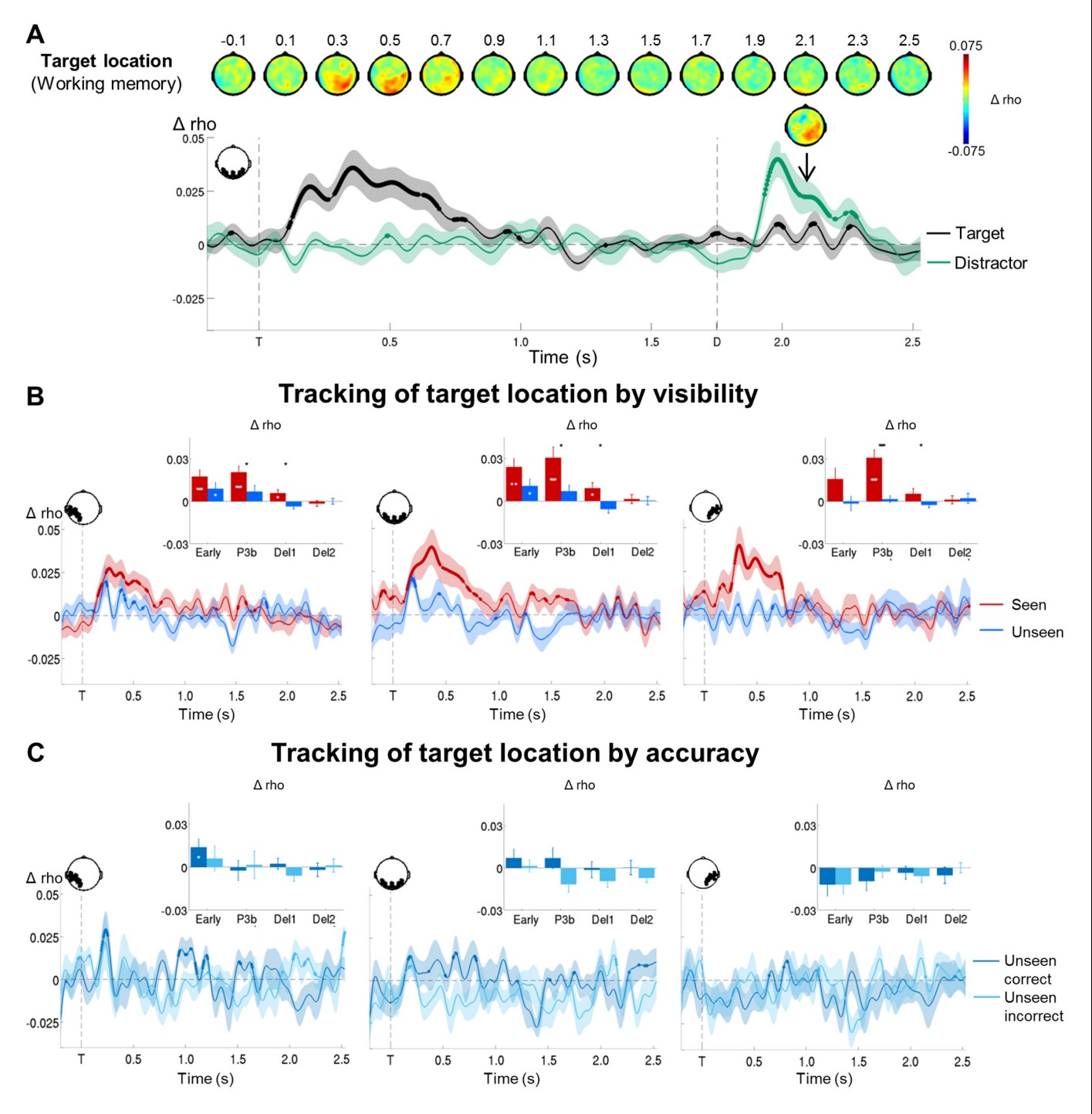

**Figure 5.** Tracking the contents of conscious and non-conscious working memory. (**A**) Topographies (top) and time courses (bottom; −0.2–2.5 s) of average circular-linear correlations between the amplitude of the MEG signal (gradiometers) and target/distractor location. Shaded area demarks standard error of the mean (SEM) across subjects. Thick line represents significant increase in correlation coefficient as compared to an empirical baseline (one-tailed Wilcoxon signed-rank test across subjects, uncorrected). (**B**) Average time courses (−0.2–2.5 s) of circular-linear correlation coefficients between amplitude of the ERFs and target location as a function of visibility in the working memory task in a group of left temporo-occipital (left), occipital (middle), and right temporo-occipital (right) gradiometers. Shaded area demarks standard error of the mean (SEM) across subjects. Thick line represents significant increase in correlation coefficient as compared to an empirical baseline (one-tailed Wilcoxon signed-rank test across subjects, uncorrected). Insets show average correlation coefficients (relative to an empirical baseline) in four time windows: 100–300 ms (early), 300–600 ms (P3b), 0.6–1.55 s (Del1), and 1.55–2.5 s (Del2). White asterisks denote significant differences to baseline (one-tailed Wilcoxon signed-rank test across subjects),

*Figure 5 continued on next page*

*Figure 5 continued*

black asterisks significant differences between conditions (two-tailed Wilcoxon signed-rank test across subjects). For display purposes, data were lowpass-filtered at 8 Hz. *p<0.05, **p<0.01, and ***p<0.001. Del1= first part of delay, Del2 = second part of delay, T = target onset. (C) Same as in (B), but as a function of accuracy on the unseen trials (correct = within ±2 positions of the target).

The following figure supplements are available for figure 5:

**Figure supplement 1.** Representation of seen target locations during conscious perception and working memory.
**Figure supplement 2.** Circular-linear correlations and multivariate decoding reveal similar time courses for target location.
**Figure supplement 3.** Tracking target/response location on unseen correct and incorrect trials with multivariate decoding.

windows: all ps<0.001; all BFs>109.60; *Figure 5A* and *Supplementary files 2* and *3*). Similarly, distractor position could be tracked between ~194 and 570 ms after its presentation (early and P3b time windows: all ps<0.009; all BFs>14.47). The position of our stimuli could thus be faithfully retrieved in visual areas.

In a subsequent step, we investigated how target location would be maintained in the context of conscious and non-conscious working memory (*Figure 5B*). Target position was transiently encoded via slowly decaying activity in occipital as well as bilateral temporo-occipital cortex from ~120 to 800 ms on seen trials (early and P3b time windows: all ps<0.001 and all BFs>24.07, with the exception of the 100–300 ms period in right temporo-occipital channels, where p=0.064 and BF=2.31) and in occipital and left temporo-occipital brain areas from ~180 to 504 ms on unseen trials (early time window: all ps<0.047; all BFs>2.58). A clear correlation with target location was therefore found for both seen and unseen trials. In fact, although it was more short-lived on the latter, it was of comparable magnitude as the one observed on the seen trials during the early time window (occipital/left temporo-occipital channels: all ps>0.110 when directly comparing the correlation scores of seen and unseen trials in a Wilcoxon signed-rank test). In the case of seen trials, both occipital and left temporo-occipital cortex also maintained the target representation at least throughout the first part of the delay period (all ps<0.024; all BFs>3.77), though, intriguingly, this was not accompanied by continuously sustained activity. Target 'decodability' instead waxed and waned, appearing and disappearing periodically. No such activity was observed for the maintenance of unseen targets (first and second part of the delay: all ps>0.446; all BFs<0.047). This absence of 'decodability' during the maintenance period persisted, even when considering unseen correct and unseen incorrect trials separately (*Figure 5C*). There was only a trace of residual decoding of target location on unseen correct trials in left temporo-occipital areas during the delay period, but this did not reach significance, potentially due to the low number of trials in this condition. Note that in the perception task, seen targets could be retrieved similarly to their counterparts in the working memory task between ~232 and 1184 ms in occipital and bilateral temporo-occipital regions (all ps>0.068, except for the 100–300 ms time window in occipital channels where p=0.008, when directly comparing the correlation scores of seen targets in both tasks in a Wilcoxon signed-rank test; *Figure 5—figure supplement 1*).

Given the univariate nature of the circular-linear correlations, one might again wonder whether a multivariate strategy would be more sensitive in detecting subtle associations between the MEG data and target location. We therefore used linear support vector regressions (SVR) to predict target angle from the MEG signal as a function of visibility (Materials and methods). As can be seen in *Figure 5—figure supplement 2*, this method resulted in similar, albeit more noisy, time courses as the ones obtained with the circular-linear correlations: Seen targets were again encoded and maintained intermittently between ~268 ms and 1.4 s (P3b time window and first part of the delay: ps<0.05). No statistically significant decoding emerged for unseen target locations. Due to the fact that subjects responded correctly on approximately half of all unseen trials (see *Supplementary file 4* for average trial counts), we attempted to evaluate the dynamics of the encoding and maintenance of unseen correct and incorrect target locations by training the regression model on the strongest case, the seen correct trials, and applying it separately to the unseen correct and incorrect trials. We again observed no evidence for any generalization at all (*Figure 5—figure supplement 3A*), though this likely reflects the sensitivity of the analysis more so than any meaningful effect.

Taken together, in line with previous research (*Harrison and Tong, 2009*; *King et al., 2016*), these results suggest that posterior sensory regions may initially encode seen and unseen memoranda via slowly decaying neural activity. In the case of conscious working memory, these then seem to be maintained by those same areas through an intermittently reactivated, neural code (*Fuentemilla et al., 2010*). In contrast, no such periodically resurfacing activity appears to accompany non-conscious working memory.

## Further evidence against the conscious maintenance hypothesis

The correlation between target location and brain activity affords an additional way to interrogate the conscious maintenance hypothesis. If subjects quickly guessed the location of an unseen target and then held it in conscious working memory, in addition to observing a signature of conscious processing on the unseen trials, we should observe a correlation with the location of their response long before it occurs. Potentially, remembering the response might recruit brain systems completely different from the ones representing the target.

Circular-linear correlations rendered this prediction unlikely. Associations between response location and the MEG signal were again primarily confined to posterior channels, with more frontal areas being recruited preferentially at the time of the response (*Figure 6A*). As such, the topographical patterns were highly similar to the ones observed for the correlation with target location. Importantly, no additional regions were identified on the unseen trials and none of these areas showed any appreciable correlation before the presentation of the response screen (*Figure 6—figure supplement 1*). This suggests that, irrespective of stimulus visibility, common brain networks supported memories for the target stimulus and the ensuing decision and that, in the case of non-conscious working memory, these did not come online until the response.

The time courses of the circular-linear correlations further solidified this interpretation (*Figure 6B*). On seen trials, response position was maintained throughout the majority of the epoch in occipital and left temporo-occipital brain areas (first three time windows: all ps<0.020; all BFs>4.16). This was not the case on the unseen trials: No correlation patterns appeared in any of the posterior channels during the course of the epoch (all time windows: all ps>0.064; all BFs<1.32). In contrast, a strong correlation emerged for both seen and unseen trials during the response period (0–800 ms with respect to the onset of the letter cue). Response location could be tracked with similar time courses and magnitude on seen and unseen trials in occipital, bilateral temporo-occipital, and frontal channels (all ps<0.024; all BFs>13.73; when directly comparing the correlation scores of seen and unseen targets in a Wilcoxon signed-rank test: all ps>0.216, except for left temporo-occipital channels, where p=0.040). When we further distinguished unseen correct from unseen incorrect trials, the results remained similar, though much noisier (*Figure 6C*): There was no clear correlation pattern before the onset of the response screen on either the unseen correct or the unseen incorrect trials (all ps>0.096; all BFs<1.47). Only after the appearance of the letter cues did we observe a correlation with response location.

Multivariate decoding analyses confirmed this picture: Whereas response location for seen targets could be tracked similarly to actual target location at least throughout the first part of the delay period (P3b time window and first part of the delay: ps<0.05; *Figure 6—figure supplement 2*), no such pattern was observed on the unseen trials (all ps>0.153). This absence of decodability persisted on the unseen correct and incorrect trials, even when training the regression model on the seen correct trials (*Figure 5—figure supplement 3B*).

Overall, these results are incompatible with the hypothesis that the long-lasting blindsight is only due to the conscious maintenance of an early guess, as, in this case, brain responses linked to the subjects' responses should have been observed shortly after the presentation of the target stimulus.

## Short-term synaptic change as a neurophysiological mechanism for conscious and non-conscious working memory

What mechanism might permit above-chance recall without any continuously sustained brain activity? Recent modelling suggests that sustained neural firing may not be required to maintain a representation in conscious working memory. *Mongillo et al. (2008)* proposed a theoretical framework for working memory, in which information is stored in calcium-mediated short-term changes in synaptic weights, thus linking the active cells coding for the memorized item. Once these changes have

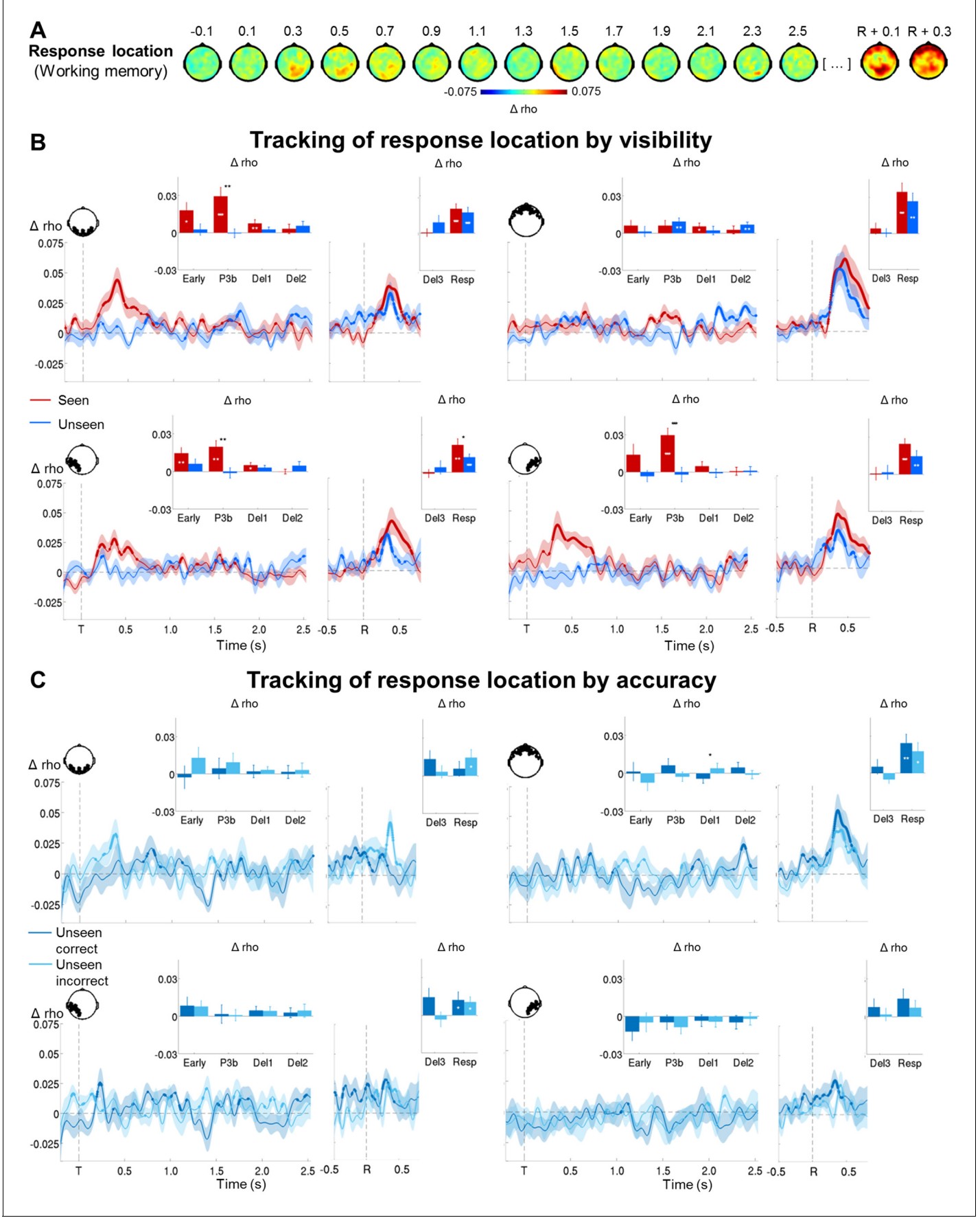

**A**
Response location
(Working memory)

**B** Tracking of response location by visibility

Seen
Unseen

**C** Tracking of response location by accuracy

Unseen correct
Unseen incorrect

**Figure 6.** Tracking response location in conscious and non-conscious working memory. (**A**) Topographies of average circular-linear correlations between the amplitude of the MEG signal (gradiometers) and response location. R = onset of the response screen. (**B**) Average time courses (left: stimulus-locked, −0.2–2.5 s; right: response-locked, −0.5–0.8 s) of circular-linear correlation coefficients between the amplitude of the ERFs and response location as a function of visibility in the working memory task in a group of occipital (top, left), frontal (top, right) left temporo-occipital (bottom, left) and right temporo-occipital (bottom, right) gradiometers. Shaded area demarks standard error of the mean (SEM) across subjects. Thick line represents significant increase in correlation coefficient as compared to an empirical baseline (one-tailed Wilcoxon signed-rank test across subjects, uncorrected). Insets show average correlation coefficients (relative to an empirical baseline) in four stimulus-locked time windows, 100–300 ms (early), 300–600 ms (P3b), 0.6–1.55 s (Del1), and 1.55–2.5 s (Del2), and two response-locked time windows, −0.5–0.0 s (Del3) and 0.0–0.8 s (Resp). White asterisks denote significant differences to baseline (one-tailed Wilcoxon signed-rank test across subjects), black asterisks significant differences between conditions (two-tailed Wilcoxon signed-rank test across subjects). For display purposes, data were lowpass-filtered at 8 Hz. *p<0.05, **p<0.01, and ***p<0.001. Del1= first part of delay, Del2 = second part of delay, Del3 = last 500 ms before response screen, R = response screen onset, T = target onset. (**C**) Same as in (**B**), but as a function of accuracy on the unseen trials (correct = within ±2 positions of the target).

The following figure supplements are available for figure 6:

**Figure supplement 1.** Topographies for circular-linear correlations with response location as a function of visibility.

**Figure supplement 2.** Circular-linear correlations and multivariate decoding reveal similar time courses for response location.

occurred, the cell assembly may go dormant during the delay, while the synaptic weights are slowly decaying. At the end of the delay period, a non-specific read-out signal may then suffice to reactivate the assembly. Furthermore, reactivation of the assembly may also occur spontaneously during the retention phase, similar to the rehearsal process postulated by *Baddeley (2003)*, thus refreshing the weights and permitting the bridging of longer delays. Could this 'activity-silent' mechanism also constitute a plausible neural mechanism for non-conscious working memory?

To test this hypothesis, we simulated our experiments using a one-dimensional recurrent continuous attractor neural network (CANN) based on *Mongillo et al. (2008)*. The CANN encoded the angular position of the target and was composed of neurons aligned according to their preferred stimulus value (*Figure 7A*). Transient short-term plasticity between the recurrent connections, with a 4s-decay constant, was implemented as described by *Mongillo et al. (2008; Figure 7B)*. Timing of the simulated events was comparable to the experimental paradigm: A target signal was briefly presented at a random location, followed by a mask signal to all neurons and a non-specific recall signal after a 3s-delay.

If the activity-silent mechanism constituted a plausible neurophysiological correlate of conscious and non-conscious working memory, these simulations should capture our principal findings. A stimulus presented at threshold should entail one of two different maintenance regimes: a first distinguished by near-perfect recall with spontaneous reactivations of the memorized representation throughout the retention period (thus resembling the prolonged, yet fluctuating, 'decodability' of seen target locations), and a second characterized by above-chance objective performance in the almost complete absence of delay activity (thereby portraying the time course of the circular-linear correlations for the unseen stimuli).

In a noiseless model, there indeed existed a critical value of mask amplitude, $A_{critical}$, which separated two distinct regimes: Just as was the case for our seen trials, when $A_{mask} < A_{critical}$, the neural assembly coding for the target spontaneously reactivated during the delay (*Figure 7C*). However, when $A_{mask} > A_{critical}$, the system evolved into a state without spontaneous activation of target-specific neurons, yet with a reactivation in response to a non-specific recall signal, mimicking our unseen trials (*Figure 7D*). When fixing mask amplitude near $A_{critical}$ and adding noise continuously or just to the inputs, the network exhibited both types of regimes in nearly equal proportions: 50.8% of trials were characterized by an activity-silent delay interspersed with spontaneous reactivations and 49.2% by an entirely activity-silent delay period. Reminiscent of our behavioral results, sorting the trials according to the existence or absence of these reactivations and computing the histograms of recalled target position relative to true location produced two distributions of objective working memory performance: one, in which target position was nearly accurately stored (*Figure 7E*), and one, in which performance remained above chance despite a higher base rate of errors (*Figure 7F*). These simulations replicate our experimental findings (in particular *Figures 2* and *5*) and suggest the

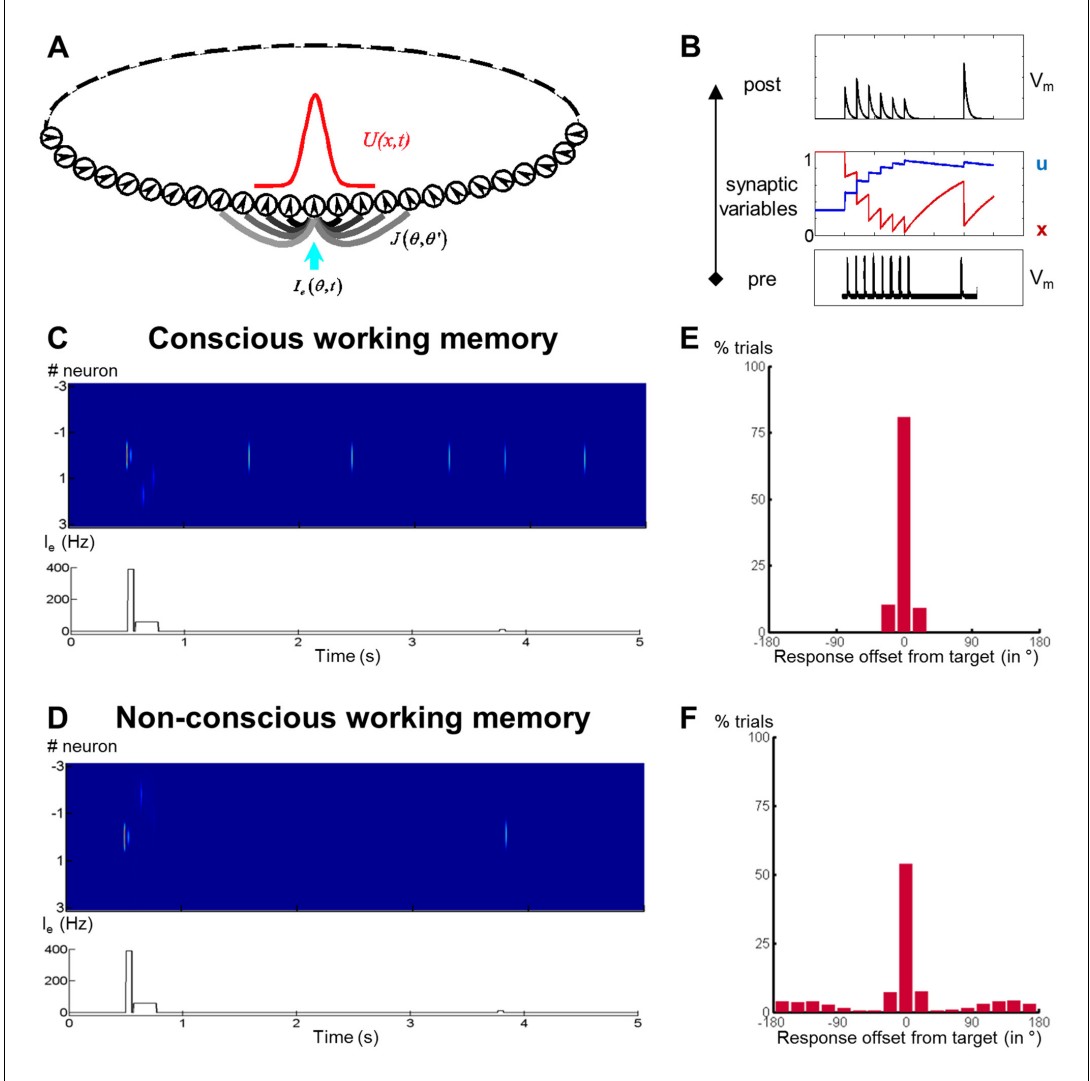

**Figure 7.** Activity-silent neural mechanisms underlying conscious and non-conscious working memory. (A) Structure of a one-dimensional continuous attractor neural network (CANN). Neuronal connections $J (\theta, \theta')$ are translation-invariant in the space of the neurons' preferred stimulus values ($-\pi$, $\pi$), allowing the network to hold a continuous family of stationary states (bumps). An external input $I_e (\theta, t)$ containing the stimulus information triggers a bump state (red curve) at the corresponding location in the network. (B) Model of a synaptic connection with short-term potentiation. In response to a presynaptic spike train (bottom), the neurotransmitter release probability $u$ increases and the fraction of available neurotransmitter $x$ decreases (middle), representing synaptic facilitation and depression. Effective synaptic efficacy is proportional to $ux$ (top). (C) Firing rate of neurons (top) and sequence of events (bottom; target and mask signal) when simulating conscious working memory with $A_{mask}$ = 50 Hz < $A_{critical}$. (D) Same as in (C) for non-conscious working memory when $A_{mask}$ = 65 Hz > $A_{critical}$. (E, F) Performance of the network (distribution of responses) when mask amplitude was near the critical level, $A_{mask}$ = 62 Hz ~ $A_{critical}$, and noise had been added to the system. Out of 4000 trials, 2035 resulted in the conscious (E) and the remainder in the non-conscious regime (F). In both cases, performance remained above chance with the responses concentrated around the initial target location.

activity-silent framework as a likely candidate mechanism for both conscious and non-conscious working memory.

## Discussion

Conscious perception and working memory are thought to be intimately related, yet recent evidence challenged this assumption by proposing the existence of non-conscious working memory (*Soto et al., 2011*). The present results may reconcile these views. Both conscious perception and conscious working memory shared similar signatures, including an alpha/beta power decrease, the

latter spanning the entire delay on working-memory trials. However, participants remained able to localize a subjectively invisible target after a 4s-delay. We found no evidence that this long-lasting blindsight could simply be explained by erroneous visibility reports or by the conscious maintenance of an early guess. It thus likely reflects genuine non-conscious working memory. Despite the inherent differences in subjective experience for conscious and non-conscious working memory, a single, activity-silent mechanism might support both conscious and non-conscious information maintenance. We now discuss these points in turn.

## Shared brain signatures underlie conscious perception and conscious working memory

Consistent with introspective reports and research on visual awareness and working memory (*Baddeley, 2003*; *Dehaene et al., 2014*), we observed a close relationship between conscious perception and maintenance in conscious working memory. In both tasks, classifiers trained to separate seen and unseen trials resulted in thick diagonals up to ~1 s after target onset, even when generalizing from one task to the other. Such long diagonals have repeatedly been observed in recent studies and are thought to reflect sequential processing (*King and Dehaene, 2014*; *Marti et al., 2015*; *Salti et al., 2015*; *Stokes et al., 2015*; *Wolff et al., 2015*). Irrespective of context, conscious perception and early parts of conscious maintenance thus involve a similar series of partially overlapping processing stages.

Time-frequency decompositions reinforced and extended this conclusion. Seen trials in the perception task were distinguished from both a target-absent control condition and unseen trials by a prominent decrease in alpha/beta power over fronto-central sensors, corresponding to a distributed network centered on parietal cortex. A similar desynchronization, sustained throughout the retention period, was also observed for conscious working memory. Alpha/beta band desynchronizations such as these have previously been linked with conscious perception (*Gaillard et al., 2009*; *Wyart and Tallon-Baudry, 2009*) and working memory (*Lundqvist et al., 2016*). Modelling suggests that the memorized item is encoded by intermittent gamma bursts, which interrupt an ongoing desynchronized beta default state (*Lundqvist et al., 2011*). Such a decreased rate of beta bursts, once averaged over many trials, would have resulted in the apparently sustained power decrease we observed. Increases in gamma power have also been shown in some studies on conscious perception (e.g., *Gaillard et al., 2009*), but we failed to detect it here, perhaps because our targets were brief, peripheral, and low in intensity.

Circular-linear correlations further highlighted the similarity between conscious perception and working memory. Location information could be tracked for ~1 s on perception-only trials and for at least 1.5 s of the working memory retention period. The mental representation formed during conscious perception was therefore either maintained or repeatedly replayed during conscious working memory.

## Long-lasting blindsight effect reflects genuine non-conscious working memory

Even when subjects indicated not having seen the target, they still identified its position much better than chance up to 4 s after its presentation. This long-lasting blindsight effect was replicated in two independent experiments and exhibited typical properties of working memory, withstanding salient visible distractors and a concurrent demand on conscious working memory. Those results corroborate previous research showing that information can be maintained non-consciously (e.g., *Bergström and Eriksson, 2014*, *2015*; *Dutta et al., 2014*; *Soto et al., 2011*). However, these prior findings could have arisen due to errors in visibility reports. If, for example, a participant had been left with a weak impression of the target (and, consequently, its location), he or she might not have had adequate internal evidence to refer to this perceptual state as seen, thus incorrectly applying the label unseen. A small number of such errors would have produced above-chance responding. Another explanation could have been the conscious maintenance of an early guess, whereby subjects would have ventured a prediction as to the correct target position immediately after its presentation and then consciously maintained this hunch.

The MEG results provide evidence against these possibilities. First, whereas seen trials were characterized by a sustained desynchronization in the alpha/beta band in parietal brain areas, no

comparable desynchronization was observed on unseen trials, even when subjects correctly identi-fied the target location. On the contrary, the only, short-lived, difference between unseen correct and unseen incorrect trials emerged around the time of the distractor and was reversed in direction: Unseen correct trials were accompanied by an increase in power in the alpha band with respect to their incorrect counterpart, an effect that might relate to a successful attempt to reduce interference from the distractor (*Cooper et al., 2003*; *Jensen and Mazaheri, 2010*). Otherwise, unseen correct and incorrect trials were indistinguishable in their power spectra and similar to the target-absent control condition. Second, there was no clear evidence for a shared discriminative decoding axis between the seen and the unseen correct trials: Generalization was entirely unsuccessful when the classifier was trained on the time-frequency data, and highly dissimilar from the original visibility decoder when trained on the ERFs. While it is impossible to draw definitive conclusions just from the current dataset and future research should replicate these results, the majority of our evidence thus points against an interpretation, in which the unseen correct trials constituted either just a subset of seen trials, or arose from the conscious maintenance of an early guess. Instead, inasmuch as the observed desynchronization serves as a faithful indicator of conscious processing, it argues in favor of a differential state of non-conscious working memory with a distinct neural signature.

Circular-linear correlations as well as multivariate regression models between the amplitude of the MEG signal and response location support this interpretation. On seen trials, response position was coded akin to target location: Initially maintained via slowly decaying neural activity in posterior brain areas, the response code subsequently resurfaced intermittently in the same as well as more frontal regions. There was no detectable evidence for such a code on the unseen trials. Only during the very last part of the delay, right before the response, did response-related neural activity emerge and ramp up to the same level as on seen trials during the response period. As such, the absence of any prior delay-period activity does not appear to be an artifact attributable to low statistical power or an increase in noise on the unseen trials. Instead, in conjunction with the absence of any signature of conscious processing on these trials, these findings imply that subjects did not consciously maintain an early guess and rather relied on genuine non-conscious working memory to perform the task.

In this context, an interesting avenue for future investigations might be to delineate the boundary conditions of such non-conscious working memory. Although the short-term maintenance of information certainly lies at the heart of most theories of working memory (*Eriksson et al., 2015*), there exist additional criteria for working memory that were not investigated in the present study. It is thus an interesting empirical question whether these other working memory processes may also occur without subjective awareness. Is it, for example, possible to manipulate information non-consciously? Though speculative, in light of the proposed activity-silent code for non-conscious maintenance (without any spontaneous reactivations; see below), it seems unlikely. Being an entirely passive process, it is not clear how stored representations could be transformed without being persistently activated and thus becoming conscious. Future research is, however, needed to provide a definitive answer.

## A theoretical framework for 'activity-silent' working memory

Target-related activity was not continuously sustained throughout the delay period, even when the target square had been consciously perceived. It instead fluctuated, disappearing and reappearing intermittently. This feature was even more pronounced on the unseen trials, with no evidence for any such retention-related activity beyond ~1 s. We presented a theoretical framework, based on *Mongillo et al. (2008)* and the concept of 'activity-silent' working memory (*Stokes, 2015*), that may provide a plausible explanation for maintenance without sustained neural activity. According to this model, short-term memories are retained by slowly decaying patterns of synaptic weights. A retrieval cue presented at the end of the delay may then serve as a non-specific read-out signal capable of reactivating these dormant representations above chance-level. Support for this model comes from experiments in which non-specific, task-irrelevant stimuli (*Wolff et al., 2017*, *2015*), neutral post-cues (*Sprague et al., 2016*), or transcranial magnetic stimulation (TMS) pulses (*Rose et al., 2016*) presented during a delay restore the decodability of representations. Direct physiological evidence for the postulated short-term changes in synaptic efficacies also exists (*Fujisawa et al., 2008*).

The present non-conscious condition provides further support for such an activity-silent mechanism. In this framework, a stimulus that fails to cross the threshold for sustained activity and subjective visibility may still induce enough activity in high-level cortical circuits to trigger short-term

synaptic changes. Such transient non-conscious propagation of activity has been simulated in neural networks (*Dehaene and Naccache, 2001*) and measured experimentally in temporo-occipital, parietal, and even prefrontal cortices (*Salti et al., 2015*; *van Gaal and Lamme, 2012*). In the present work, we indeed observed some residual, transiently decodable activity over left temporo-occipital sensors on unseen correct trials. The memory of target location could therefore have arisen from posterior visual maps (*Roelfsema, 2015*), although future research should test this prediction further. Note that activity-silent mechanisms need not apply solely to prefrontal cortex as originally proposed by *Mongillo et al. (2008)*, but constitute a generic mechanism that may be replicated in different areas, possibly with increasingly longer time constants across the cortical hierarchy (*Chaudhuri et al., 2014*). Only some of these areas/spatial maps may be storing the information on unseen trials.

A key feature of the model by *Mongillo et al. (2008)* and the present simulations is that, even for above-threshold ('seen') stimuli, delay activity is not continuously sustained. Occasional bouts of spontaneous reactivation instead refresh the synaptic weights and maintain the memory for an indefinite time. The time courses of the circular-linear correlations and of the multivariate decoding we observed on seen trials match this description: While target location was encoded and maintained in temporo-occipital areas, target 'decodability' was not constantly sustained, but waxed and waned throughout the delay. *Fuentemilla et al. (2010)* also observed that, during a delay period, decodable representations of memorized images recurred at a theta rhythm. More recently, single-trial analyses of monkey electrophysiological recordings in a working memory task have confirmed the absence of any continuous activity and instead identified the presence of discrete gamma bursts, paired with a decrease in beta-burst probability (*Lundqvist et al., 2016*). Such periodic refreshing of otherwise activity-silent representations could potentially serve as the neural correlate of conscious rehearsal, a central feature of working memory according to *Baddeley (2003)*. It also suggests, however, that even consciously perceived items may not always be 'in mind.' Future research might attempt to more directly simulate activity-silent mechanisms in the context of conscious and non-conscious perception by, for example, relying on more elaborate models capturing decreases in alpha/beta power (*Lundqvist et al., 2011*).

In conjunction with prior evidence (*King et al., 2016*; *Salti et al., 2015*), our findings therefore indicate that there may be two successive mechanisms for the short-term maintenance of conscious and non-conscious stimuli: an initial, transient period of ~1 s, during which the representation is encoded by active firing with a slowly decaying amplitude, and an ensuing activity-silent maintenance via short-term changes in synaptic weights, during which activity either intermittently resurfaces (conscious case) or vanishes (non-conscious case). Such activity-silent retention need not necessarily be specific to working memory. Recent investigations have, for instance, demonstrated the existence of recognition memory for invisible cues (*Chong et al., 2014*; *Rosenthal et al., 2016*). As delay periods ranged in the order of minutes rather than seconds, persistent neural activity seems to be an unlikely candidate mechanism of maintenance. Activity-silent codes might have been at play, though they probably depended on mechanisms with longer time constants than the relatively rapidly decaying patterns of synaptic weights discussed in the context of the present experiments. Nevertheless, activity-silent representations may constitute a general mechanism for maintenance across the whole spectrum of temporal delays (from seconds over minutes/hours to days/weeks/decades), thus forming a generic property of memory.

## Limitations and future perspectives

Our study presents limitations that should be addressed by future research. Due to the nature of the current investigation (a working memory task with long trials and subjectively determined variables), a relatively small number of unseen trials was acquired, thus making it difficult to detect subtle effects. While our conclusions are supported by Bayes Factor analyses, converging evidence from univariate and multivariate techniques, and similar results obtained with larger samples in the domain of activity-silent conscious working memory (e.g., *Rose et al., 2016*; *Wolff et al., 2017*), a number of our observations are based on null effects, and it remains a possibility that we missed some target- and/or response-related activity on the unseen trials. Future research should thus aim at replicating the present findings with larger datasets or with more sensitive techniques, such as intra-cranial recordings. In particular, it might be interesting to further probe the relationship between seen, unseen correct, and unseen incorrect targets: A specific prediction of the proposed

model is that unseen correct trials should possess enough activity to modify synaptic weights in high-level cortical circuits, yet without crossing the threshold for sustained activity and consciousness ('failed ignition'). Unseen correct trials should thus share some of the processes that are found on seen trials and future research is necessary to directly test this hypothesis.

## Conclusion

In contrast to a widely-held belief, our findings support the existence of genuine working memory in the absence of either conscious perception or sustained activity. Our proposal is that, following a transient encoding phase via active firing, non-conscious stimuli may be maintained by 'activity-silent' short-term changes in synaptic weights without any detectable neural activity, allowing above-chance retrieval for several seconds. Similar activity-silent codes also subserve conscious maintenance, though in this case periodic refreshing appears to stabilize the stored representations throughout the delay. Our findings thus highlight the need to refine our understanding of working memory, and to continuously challenge the limits of non-conscious processing.

# Materials and methods

## Subjects

38 healthy volunteers participated in the present study (experiment 1: N = 17, $M_{age}$ = 23.3 years, $SD_{age}$ = 2.8 years, 10 men; experiment 2: N = 21, $M_{age}$ = 24.3 years, $SD_{age}$ = 3.8 years, 9 men). They gave written informed consent and received 80 or 15€ as compensation for the imaging and behavioral paradigms. Due to noisy recordings, only 13 of the 17 subjects in experiment 1 were retained for the MEG analyses. Although sample size had not specifically been estimated for our study, it thus was reasonable given typical experiments in the field.

## Experimental protocol

Participants performed variations of a spatial delayed-response task, designed to assess the retention of a target location under varying levels of subjective visibility (*Figure 1A*). Each trial began with the presentation of a central fixation cross (500 ms), displayed in white ink on an otherwise black screen. In experiment 1, a faint gray target square (RGB: 89.25 89.25 89.25) was flashed for 17 ms in 1 out of 20 equally spaced, invisible positions along a circle centered on fixation (radius = 200 pixels; eight repetitions/location). Another fixation cross (17 ms) preceded the display of the mask (233 ms). Mask elements were composed of four individual squares (two right above and below, and two to the left and right of the target stimulus), arranged to tightly surround the target square without overlapping it. They appeared simultaneously at all possible target locations. Mask contrast was adjusted on an individual basis in a separate calibration procedure (see below). A variable delay period with constant fixation followed the mask (2.5, 3.0, 3.5, or 4.0 s). On 50% of the trials in experiment 1, an unmasked distractor square, randomly placed and with the same duration as the target, was presented 1.5 s into the delay period.

After the delay, 20 letters – drawn from a subset of lower-case letters of the alphabet (excluded: *e, j, n, p, t, v*) – were randomly presented in the 20 positions (2.5 s). Participants were asked to identify the target location by speaking the name of the letter presented at the location. They were instructed to always provide a response, guessing if necessary. A trial ended with the presentation of the word *Vu?* (French for *seen*) in the center of the screen (2.5 s), cueing participants to rate the visibility of the target on the 4-point Perceptual Awareness Scale (PAS; *1*: no experience of the target, *2*: brief glimpse, *3*: almost clear experience, *4*: clear experience; *Ramsøy and Overgaard, 2004*) using the index, middle, ring, or little finger of their right hand (five-button non-magnetic response box, Cambridge Research Systems Ltd., Fiber Optic Response Pad). We instructed subjects to reserve a visibility rating of *1* for those trials, for which they had absolutely no perception of the target. The target square was also replaced by a blank screen on 20% of the trials, in order to obtain an objective measure of participants' sensitivity to the presence of the target. The inter-trial interval (ITI) lasted 1 s. Subjects completed a total of 200 trials of this working memory task, divided into four separate experimental blocks. They also undertook two blocks of 100 trials each of a perception-only control paradigm, identical to the working memory task in all respects except that the delay period and target localization screen were omitted, such that the presentation of the mask

immediately preceded subjects' visibility ratings. Task order (perception vs. working memory) was counterbalanced across participants.

Experiment 2 was designed to investigate the impact of a conscious working memory load on non-conscious working memory. Apart from the following exceptions, it was identical to experiment 1: A screen with either 1 (low load) or 5 (high load) centrally presented digits (1.5 s) – randomly drawn (without replacement) from the numbers 1 through 9 – as well as a 1s-fixation period were shown prior to the presentation of the target square. Following either a 0s- or a 4s-delay period, subjects first identified the target location by typing their responses on a standard AZERTY keyboard (4 s). The French word for *numbers* (*Numéros?*) then probed participants to recall the sequence of digits in the correct order. Responses were again logged on the keyboard during a period of 4.5 s. Subjects last rated target visibility as in experiment 1 (3 s). The ITI varied between 1 and 2 s. Participants completed two experimental blocks of 100 trials each.

## Calibration task

Prior to the experimental tasks, each participant's perceptual threshold was estimated in order to ensure roughly equal proportions of seen and unseen trials. Subjects completed 150 (experiment 1: three blocks) or 125 (experiment 2: five blocks) trials of a modified version of the working memory task (no distractor, delay duration: 2 s in experiment 1 and 0 s in experiment 2), during which mask contrast was either increased (following a visibility rating of *2*, *3*, or *4*) or decreased (following a visibility rating of *1*) on each target-present trial according to a double-staircase procedure. Individual perceptual thresholds to be used in the main tasks were derived by averaging the mask contrasts from the last four switches from seen to unseen (or vice versa) of each staircase.

## Behavioral analyses

We analyzed our behavioral data in Matlab R2014a (MathWorks Inc., Natick, MA; code available upon request) and SPSS Statistics Version 20.0 (IBM, Armonk, NY), using repeated-measures analyses of variance (ANOVAs). Only meaningful trials without missing responses were included in any analysis. Distributions of localization responses were computed for visibility categories with at least five trials per subject. Objective working memory performance was quantified via two complementary measures. The *rate of correct responding* was defined as the proportion of trials within two positions (i.e., ±36°) of the actual target location and served as an index of the amount of information that could be retained. Because 5 out of 20 locations were counted as correct, chance on this measure was 25%. The *precision* of working memory was estimated as the dispersion (standard deviation) of spatial responses. In particular, we modeled the observed distribution of responses *D(n)* as a mixture of a uniform distribution (random guessing) and an unknown probability distribution *d* ('true working memory'):

$$D(n) = \frac{p}{N} + (1-p)d(n) \tag{1}$$

where *p* refers to the probability that a given trial is responded to using random guessing; *N* to the number of target locations (*N* = 20); and *n* is the deviation from the true target location. We assumed that *d(n)* = 0 for deviations beyond a fixed limit *a* (with *a* = 2). This hypothesis allowed us to estimate *p* from the mean of that part of the distribution *D* for which one may safely assume no contribution of working memory:

$$\hat{p} = \frac{\sum D(n) | n \, \text{outside} [-a, a]}{(N - 2a - 1)} * N \tag{2}$$

where the model is designed in such a way as to ensure that $\hat{p} = 1$ if *D* is a uniform distribution (i.e., 100% of random guessing) and $\hat{p} = 0$ if *D* vanishes outside the region of correct responding (i.e., 0% of random guessing). There needs to be at least chance performance inside the region of correct responding, so

$$\sum D(n) | n \epsilon [-a, a] \geq \frac{[2a - 1]}{N} \tag{3}$$

which ensures $0 \leq \hat{p} \leq 1$. This is the reason why, when computing precision, we included only subjects

whose rate of correct responding for unseen trials, collapsed across all experimental conditions, significantly exceeded chance performance (i.e., 25%) in a $\chi^2$-test (p<0.05). An estimate of $d$, $\hat{d}$, can then be derived in two steps from *Equation 1* as

$$\delta(n) = \frac{D(n) - \frac{\hat{p}}{N}}{1 - \hat{p}} \tag{4}$$

$$\hat{d}(n) = \frac{\delta(n)|n\epsilon[-a, a]}{\sum \delta(n)|n\epsilon[-a, a]} \tag{5}$$

We note that the distribution $\delta$ has residual, yet negligible, positive and negative mass (due to noise) outside the region of correct responding. In order to obtain $\hat{d}$, we therefore restricted the distribution $\delta$ to $[-a, a]$, set all negative values to 0, and renormalized its mass to 1. The precision of the representation of the target location in working memory was then defined as the standard deviation of that distribution.

## MEG recordings and preprocessing

In experiment 1, we recorded MEG with a 306-channel (102 sensor triplets: 1 magnetometer and 2 orthogonal planar gradiometers), whole-head setup by ElektaNeuromag (Helsinki, Finland) at 1000 Hz with a hardware bandpass filter between 0.1 and 330 Hz. Eye movements as well as heart rate were monitored with vertical and horizontal EOG and ECG channels. Prior to installation of the subject in the MEG chamber, we digitized three head landmarks (nasion and pre-auricular points), four head position indicator (HPI) coils placed over frontal and mastoïdian skull areas, and 60 additional locations outlining the participant's head with a 3-dimensional Fastrak system (Polhemus, USA). Head position was measured at the beginning of each run.

Our preprocessing pipeline followed *Marti et al. (2015)*. Using MaxFilter Software (ElektaNeuromag, Helsinki, Finland), raw MEG signals were first cleaned of head movements, bad channels, and magnetic interference originating from outside the MEG helmet (*Taulu et al., 2004*), and then downsampled to 250 Hz. We conducted all further preprocessing steps with the Fieldtrip toolbox (http://www.fieldtriptoolbox.org/; *Oostenveld et al., 2011*) run in a Matlab R2014a environment. Initially, MEG data were epoched between −0.5 and +2.5 s with respect to target onset for all stimulus-locked, and between −0.5 and +0.8 s with respect to the onset of the response screen for all response-locked analyses. Trials contaminated by muscle or other movement artifacts were then identified and rejected in a semi-automated procedure, for which the variance of the MEG signals across sensors served as an index of contamination. To remove any residual eye-movement and cardiac artifacts, we performed independent component analysis separately for each channel type, visually inspected the topographies and time courses of the first 30 components, and subtracted any contaminated component from the MEG data. Except for analyses requiring higher spatial precision (i.e., circular-linear correlations and decoding), results are presented for magnetometers only.

Further preprocessing steps depended on the nature of the subsequent analysis: Epochs retained for investigations based on evoked responses (i.e., ERFs, decoding, circular-linear correlations) were low-pass filtered at 30 Hz, while time-frequency decompositions relied on entirely unfiltered data. In the latter case, a sliding, frequency-independent Hann taper (window size: 500 ms, step size: 20 ms) was convolved with the unfiltered epochs in order to extract an estimate of power between 1 and 99 Hz (in 2 Hz steps) to identify the neural correlates of conscious and non-conscious perception and working memory in the frequency domain. Prior to univariate or multivariate statistical analysis, data (ERFs, time-frequency power estimates) were baseline corrected using a period between −200 and −50 ms.

## Circular-linear correlations

To localize and track the neural representations of target, response, and distractor location, filtered epochs were transformed into circular-linear correlation coefficients. Following *King et al. (2016)*, we combined the two linear correlation coefficients between the MEG signal and the sine and cosine of the angle defining the location in question (i.e., target, distractor, or response). An empirical null distribution was generated for each condition separately by shuffling the labels (i.e., target,

distractor, or response location) at the corresponding time points and averaging the resulting distribution from 1000 such permutations.

Due to the spatial nature of our task, there is a possibility that subjects could have systematically moved their eyes after the presentation of the target, thus contaminating the correlation analyses. However, several lines of evidence suggest that this was not the case: First, participants were carefully instructed not to move their eyes. A close inspection of the EOG traces confirmed that subjects successfully implemented this request and did not display any strategic eye movements. Second, we carefully removed any trials contaminated by such movements as part of our preprocessing procedure. Third, the topographical patterns of the correlations show that the signal primarily originated in occipital and parietal channels. Eye movements therefore unlikely have driven the circular-linear correlations.

## Sources

Individual anatomical magnetic resonance images (MRI), obtained with a 3D T1-weighted spoiled gradient recalled pulse sequence (voxel size: 1 * 1 * 1.1 mm; repetition time [TR]: 2300 ms; echo time [TE]: 2.98 ms; field of view [FOV]: 256 * 240 * 176 mm; 160 slices) in a 3T Tim Trio Siemens scanner, were first segmented into gray/white matter as well as subcortical structures with FreeSurfer (https://surfer.nmr.mgh.harvard.edu/). We then reconstructed the cortical, scalp, and head surfaces in Brainstorm (http://neuroimage.usc.edu/brainstorm; *Tadel et al., 2011*) and co-registered these anatomical images with the MEG signals, using the HPI coils and the digitized head shape as a reference. Current density distributions on the cortical surface were subsequently estimated separately for each condition and subject. Specifically, we employed an analytical model with overlapping spheres to compute the leadfield matrix and modeled neuronal current sources with an unconstrained (dipole orientation loosening factor: 0.2) weighted minimum-norm current estimate (wMNE; depth-weighting factor: 0.5) and a noise covariance obtained from the baseline period of all trials. Average time-frequency power in the alpha (8–12 Hz) and beta (13–30 Hz) bands was then estimated with complex Morlet wavelets using the Brainstorm default parameters, the resulting transformations projected onto the ICBM 152 anatomical template (*Fonov et al., 2011*, *2009*), and the contrasts between the conditions of interest computed. Group averages for spatial clusters of at least 150 vertices are shown in dB relative to baseline and were thresholded at 60% of the maximum amplitude (cortex smoothed at 60%).

## Multivariate pattern analyses

We employed the Scikit-Learn package (*Pedregosa et al., 2011*) as implemented in MNE 0.13 (*Gramfort et al., 2013*, *2014*) in order to conduct our multivariate pattern analyses (MVPA). Following *Marti et al. (2015)* and *King et al. (2016)*, we fit linear estimators at each time sample within each participant to isolate the topographical patterns best differentiating our experimental conditions. Support vector machines (*Chang and Lin, 2011*) were trained in the case of categorical data (i.e., visibility/accuracy) and a combination of two linear support vector regressions was used for circular data (i.e., target/response location) to estimate an angle from the arctangent of the separately predicted sine and cosine of the labels of interest.

A 5- (for categorical variables) or, due to the much larger number of labels, 2-fold (for circular variables), stratified cross-validation procedure was used in order to avoid overfitting: MEG data were first split into five (two) sets of trials with the same proportion of samples for each class. Within each fold, four (one) of these sets served as the training data and the remainder as the testing data. Model fitting, including all preprocessing steps, was exclusively performed on the training set. 50% of the most informative features (i.e., channels) were selected by means of a simple, univariate analysis of variance to reduce the dimensionality of the data (*Charles et al., 2014*; *Haynes and Rees, 2006*), the remaining channel-time features z-score normalized, and a weighting procedure applied in order to counteract the effects of any class imbalances. The classifier was then trained on the resulting data and applied to the left-out trials in order to identify the hyperplane (i.e., topography) best suited to separate the classes. This sequence of events (univariate feature selection, normalization, training and testing) was repeated five (two) times, ensuring that each trial would be included in the test set once.

Within the same cross-validation loop, we also evaluated the ability of each classifier to discriminate the experimental conditions of interest at all other time samples (i.e., generalization across time). This kind of MVPA results in a temporal generalization matrix, in which each entry represents the decoding performance of each classifier trained at time point t and tested at time point t', and in which the diagonal corresponds to classifiers trained and tested on the same time points (*King and Dehaene, 2014*). Importantly, when interrogating the capacity of our classifiers to generalize across tasks or labels (e.g., from the perception to the working memory task, or from seen to unseen correct target locations), we modified the aforementioned cross-validation procedure to capitalize on the independence of our training and testing data (see http://martinos.org/mne/dev/auto_examples/decoding/plot_decoding_time_generalization_conditions.html#example-decoding-plot-decoding-time-generalization-conditions-py). As such, classifiers from each training set were directly applied to the entire testing set and the respective predictions averaged.

Classifiers for categorical data generated a continuous output in the form of the distance between the respective sample and the separating hyperplane for each test trial. In order to be able to compare classification performance across subjects, we then applied a receiver operating characteristic analysis across trials within each participant and summarized overall effect sizes with the area under the curve (AUC). Unlike average decoding accuracy, the AUC serves as an unbiased measure of decoding performance as it represents the true-positive rate (e.g., a trial was correctly categorized as seen) as a function of the false-positive rate (e.g., a trial was incorrectly categorized as seen). Chance performance, corresponding to equal proportions of true and false positives, therefore leads to an AUC of 0.5. Any value greater than this critical level implies better-than-chance performance, with an AUC of 1 indicating a perfect prediction for any given class. In contrast, classifiers for circular data were first summarized by computing the mean absolute difference between the predicted and the actual angle (range: 0 to $\pi$; chance: $\pi/2$) and then transformed into an 'accuracy' score (range: $-\pi/2$ to $\pi/2$; chance: 0). To facilitate comparability between different conditions, an additional baseline-correction was then performed.

## Statistical analyses

We performed statistical analyses across subjects. For the ERF and time-frequency data, cluster-based, non-parametric t-tests with Monte Carlo permutations were used to identify significant differences between experimental conditions (*Maris and Oostenveld, 2007*). Further planned comparisons of ERF time courses (seen vs. unseen) in a-priori defined spatio-temporal regions of interest (i. e., P3b time window: 300–600 ms) were conducted with non-parametric signed-rank tests ($p_{uncorrected} < 0.05$). A correction for multiple comparisons was then applied with a false discovery rate ($p_{FDR} < 0.05$).

Non-parametric signed-rank tests ($p_{uncorrected} < 0.05$) were also employed to evaluate decoding performance and the strength of circular-linear correlations. Specifically, we assessed whether classifiers could predict the trials' classes better than chance (categorical data: AUC > 0.5; circular data: rad > 0) and whether circular-linear correlation coefficients deviated from an empirical baseline ($\Delta$rho > 0). We report temporal averages over four a-priori time bins, corresponding to an early perceptual period (100–300 ms), the P3b time window (300–600 ms), and the first (0.6–1.55 s) and second (1.55–2.53 s) part of the delay period. To capitalize on the increased spatial selectivity of gradiometers, averaged time courses of these two channels are shown for circular-linear correlations.

Bayesian statistics, based on either two- (time-frequency analyses) or one-sided (circular-linear correlations) t-tests, were also computed when appropriate with a scale factor of r = 0.707 (*Rouder et al., 2009*).

## Simulations

A one-dimensional, recurrent continuous attractor neural network (CANN) model (*Mongillo et al., 2008*) was adapted in order to simulate the experimental findings (*Figure 7A*). Individual neurons were aligned according to their preferred stimulus value, enabling the network to encode angular position of a target stimulus (range: $-\pi$ to $\pi$; periodic boundary condition). The dynamics of this system were determined by the synaptic currents of each neuron given by

$$\tau\frac{\partial h_E(\theta,t)}{\partial t} = -h_\theta + \rho \oint_{-\pi}^{\pi} J(\theta,\theta')U(\theta',t)X(\theta',t)R_E(\theta',t)d\theta' - J_{EI}R_I + I_b + \delta_1\xi_1(\theta,t) + I_e + \delta_2\xi_2(\theta,t), \quad (6)$$

$$\frac{\partial u(\theta,t)}{\partial t} = \frac{U - u(\theta,t)}{\tau_f} + U[1 - u(\theta,t)]R_E(\theta,t), \quad (7)$$

$$\frac{\partial x(\theta,t)}{\partial t} = \frac{1 - x(\theta,t)}{\tau_d} - u(\theta,t)X(\theta,t)R_E(\theta,t), \text{and} \quad (8)$$

$$\tau\frac{\partial h_I}{\partial t} = -h_1 + J_{IE}\int_{-\pi}^{\pi} R_E(\theta,t), \quad (9)$$

where $\tau$ describes the time constant of firing rate dynamics (in the order of milliseconds); $\rho$ refers to neuronal density; $h_E(\theta,t)$ and $R_E(\theta,t)$ capture the synaptic current to and firing rate of neurons with preference $\theta$ at time $t$ respectively; and $R(h) = \alpha \ln(1 + \exp(h/\alpha))$ is the neural gain chosen in the form of a smoothed threshold-linear function. $J_{IE}$ and $J_{EI}$ represent the connection strength between excitatory and inhibitory neurons. All excitatory neurons received a constant background input, $I_e$, reflecting the arousal signal when the neural system was engaged in a working memory task. $\delta_1\xi_1$ is background noise; $I_e$, any external stimulus (e.g., target, mask, and recall signal); and $\delta_1\xi_1(t)$ the noise related to those external stimuli. $u(\theta,t)$ and $x(\theta,t)$ denote the short-term synaptic facilitation (STF) and depression (STD) effects at time $t$ of neurons with preference $\theta$, respectively. The short-term plasticity dynamics are characterized by the following parameters: $J_1$ (absolute efficacy), $U$ (increment of the release probability when a spike arrives), $\tau_f$ and $\tau_d$ (facilitation and depression time constants). The STF value $u(\theta,t)$ is facilitated whenever a spike arrives, and decays to the baseline $U$ within the time $\tau_f$. The neurotransmitter value $x(\theta,t)$ is utilized by each spike in proportion to $u(\theta,t)$ and then recovers to its baseline, 1, within the time $\tau_d$.

$J(\theta,\theta')$ is the interaction strength from neurons at $\theta$ to neurons at $\theta'$ and is chosen to be

$$J(\theta,\theta') = \begin{cases} J_1\cos[B*(\theta-\theta')] - J_0 & \text{if } B*(\theta-\theta')\epsilon[-\arccos(-J_0/J_1), \arccos(-J_0/J_1)], \\ -J_0, & \text{else} \end{cases} \quad (10)$$

where $J_0$, $J_1$, and $B$ are constants which determine the connection strength between the neurons. Note that $J(\theta,\theta')$ is a function of $\theta - \theta'$, i. e., the neuronal interactions are translation-invariant in the space of neural preferred stimuli. The other parameters of the system were as follows: $\tau = 0.008$ s, $\tau_f = 4$ s, $\tau_d = 0.3$ s, $J_1 = 12$, $J_0 = 1$, $J_{EI} = 1.9$, $J_{IE} = 1.8$, $I_b = -0.1$ Hz, $\delta_1 = 0.3$, $\delta_2 = 9$, $N = 100$, $\alpha = 1.5$, $B = 2.2$.

During our simulations, we first presented a target signal with an amplitude of $A_{target} = 390$ Hz at a random location (50 ms), waited for 17 ms, and then applied a mask signal to all the neurons in the system (200 ms). The amplitude of the mask signal was initially varied in order to determine a critical value which would produce two distinct maintenance patterns, but was then fixed at a threshold of $A_{mask} = 62$ Hz. At the end of a 3s-delay period, a non-specific recall signal was given for 50 ms with $A_{recall} = 10$ Hz. Remembered target position was calculated as the population vector angle during this time period.

## Acknowledgements

We gratefully acknowledge Henrik Ueberschär, Leila Azizi, and Virginie Van Wassenhove for their invaluable daily support and stimulating discussion.

## Additional information

### Funding

| Funder | Grant reference number | Author |
|--------|------------------------|--------|
| Ecole des Neurosciences de | PhD Fellowship | Darinka Trübutschek |

| | | |
|---|---|---|
| Paris | | |
| Fondation Schneider Electric | PhD Fellowship | Darinka Trübutschek |
| CEA | | Stanislas Dehaene |
| Institut National de la Santé et de la Recherche Médicale | | Stanislas Dehaene |
| Collège de France | | Stanislas Dehaene |
| European Research Council | Senior Grant, NeuroConsc | Stanislas Dehaene |
| Fondation Roger de Spoelberch | | Stanislas Dehaene |
| Canadian Institute for Advanced Research | | Stanislas Dehaene |

The funders had no role in study design, data collection and interpretation, or the decision to submit the work for publication.

### Author contributions
DT, Conceptualization, Data curation, Formal analysis, Funding acquisition, Visualization, Methodology, Writing—original draft, Project administration, Writing—review and editing; SM, Conceptualization, Formal analysis, Supervision, Visualization, Writing—review and editing; AO, Conceptualization, Formal analysis, Methodology; J-RK, Conceptualization, Methodology, Writing—review and editing; YM, MT, Methodology, Writing—review and editing; SD, Conceptualization, Resources, Supervision, Funding acquisition, Writing—review and editing

### Author ORCIDs
Darinka Trübutschek, http://orcid.org/0000-0001-7977-1366

### Ethics
Human subjects: The study was approved by the by CPP IDF under the reference CPP 08 021. All subjects gave written informed consent and consent to publish before participating in the study.

## Additional files

### Supplementary files
• Supplementary file 1.

• Supplementary file 2.

• Supplementary file 3.

• Supplementary file 4.

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
