## [Decision Letter]

Thank you for submitting your article "A theory of working memory without consciousness or sustained activity" for consideration by *eLife*. Your article has been reviewed by two peer reviewers, and the evaluation has been overseen by a Reviewing Editor and Sabine Kastner as the Senior Editor. The reviewers have opted to remain anonymous.

The reviewers have discussed the reviews with one another and the Reviewing Editor has drafted this decision to help you prepare a revised submission.

Summary:

The manuscript was well received by both reviewers. They were impressed with several aspects of the work, including the timely topic, the novelty of experimental tests, state-of-the-art methods and excellent writing. However, they raised a number of issues that must be addressed before the manuscript will be considered for publication in *eLife*. These are summarized below and laid out in more detail in individual reviews.

Essential revisions:

1) Please clarify how a disassociated pattern of MEG responses can account for the issue related to errors in visibility reporting (i.e.,trials misclassified as unseen).

2) Address the question raised by reviewer 1 whether cue detection quantified with d' correlates with working memory performance.

3) Please discuss if the "activity-silent" code represents a more general property or is specific to non-conscious working memory as well as discuss what types of WM processes are more likely dependent on consciousness.

4) Address the problem of the accuracy on trials reported as unseen (i.e., trials misclassified as unseen), raised by reviewer 2 ("Interpretation 1").

5) Address a possibility, raised by reviewer 2, that subjects retain the "blindsight" decision to the end of the trial ("Interpretation 2") and that memory for the event and for the decision may have different neural markers.

6) Clarify the reasons for referring to extended duration "blindsight" as working memory (reviewer 2, "Interpretation 3").

7) Please address the issue of statistical power raised by reviewer 2 ("Methodology 1"). Specifically, the power of analysis is limited by the small number of error trials.

8) For decoding, use a classifier trained on "seen" vs "unseen" trials to distinguish between "unseen correct "vs "unseen incorrect" (reviewer 2, "Methodology 2")

Reviewer #1:

The theoretical basis for this study is strong and the analytical methods are very sound. There have been recent reports of 'non-conscious' WM that have been controversial and subject to alternative accounts. The present work does an excellent job in devising novel experimental tests that provide the strongest behavioural effects of 'non-conscious' WM to date. Most crucially the MEG analyses appear to rule out some alternative accounts that have been put forward against the phenomenon on 'non-conscious' WM based on 'erroneous visibility reports' and 'conscious guessing of a non-conscious target'. Furthermore, the MEG data shows that 'non-conscious' WM is associated with neural markers dissociated from those found in conscious WM in terms of (reversed) increase in alpha power and alpha/beta desynchronization during the memory delay, respectively. Target position could be decoded early on during the trial from visual cortex in both conscious and non-conscious WM. However target position could only be decoded in temporal cortex during the maintenance phase of the conscious WM trials, but not in the non-conscious trials. This absence of decodability of target location in the non-conscious trials during the maintenance phase alongside computer simulation results that mimic the behavioral data is taken to reflect an 'activity-silent' neural coding in non-conscious WM.

Comments

It is repeatedly stated that prior findings on non-conscious WM could have arisen due to errors in reporting (i.e. reporting unseen instead of partially seen), which could explain above chance performance. However it is not clear whether the same could not apply to the present study. I am unclear as to how a dissociated pattern of MEG responses may inform this. Could the authors clarify it further?

The authors may also justify why a subjective measure of awareness was used instead of an objective criterion of lack of awareness e.g. cue detection d' = 0

What is the cue detection d'? This could be computed using the false alarms on the 20% catch trials in which targets were seen. This does not appear to be reported. Is cue detection d' correlated with individual WM performance?

This absence of decodability of target location during the memory delay in the non-conscious trials is very intriguing and the computer simulations are very appropriate. They suggest that target information is sustained in an activity-silent neural code. A series of recent studies (Rosenthal et al., 2016; Chong et al., 2014) have also demonstrated recognition memory for unaware cues associated with null cue detection d'. In these studies, the learning and the test phase are much further apart, even in the order of minutes rather than seconds. In this scenario, persistent neural activity for memory targets is extremely unlikely. The authors could also discuss their findings in light of this work: is the proposed 'activity-silent' code specific to non-conscious WM or does it represent a more general property of memory including longer-term recognition memory.

If WM can be decoupled from conscious awareness, I understand this would cast doubt on the construct of WM itself; or is it just that only a subset of WM processes/tasks may be divorced from awareness? What types of WM processes do the authors think are more dependent on consciousness? I believe that some further discussion along the lines would increase the impact of the paper and its access to a general readership.

References

Rosenthal, C.R. et al., (2016). Learning and recognition of a non-conscious sequence of events in human primary visual cortex, Current Biology, 26, 834-841

Chong, T.T-J., Husain, M., and Rosenthal, C.R. (2014). Recognizing the unconscious. Current Biology, 24, R1033-1035

Reviewer #2:

Trubutschek et al. present the results of an MEG experiment and network model simulations to argue that WM does not depend on consciousness or persistent delay activity. Specifically, they show behaviourally that participants can respond above chance in a spatial working memory task, despite reporting no conscious experience of the memory item. In the brain, they show that non-conscious WM lacks two key neural signatures: sustained desynchronization of fronto-central beta and item-location decodability (but see middle panel of Figure 5). Finally, they report simulation results from a model adapted from Mongillo et al., (2008) to show that even in the absence of periodic refreshing (which could give rise to apparent sustained decoding when averaged across trials), changes in synaptic strength can still persist sufficiently long to guide above-chance guessing at the time-scale examined here (if I understand correctly). Although I very much enjoyed this paper (interesting question, state-of-the-art methods and well -written), I was left with a number of concerns regarding both theoretical interpretation and methodological details.

1) Interpretation 1. As I understand the logic, previous evidence for 'unconscious' working memory could be an artefact of conscious report errors. If we can assume that people do sometimes make errors in their subjective report (seems likely, even if that just means pressing the wrong button on some trials), then there should exist a subset of trials in which the subject says unseen, but still has conscious working memory (and so therefore can localise the item just fine). Of course, there will also be trials in which the subject accidentally reports the wrong location, but so long as these two types of response error are not perfectly correlated, then there should always exist a subset of trials such that: seen (but erroneously responded unseen) + conscious WM (faithfully reported), just as there should also be cases in which Ps erroneously report seen, but actually have no useful location in mind (i.e., guesses in the 2-4 visibility conditions). You only need to assume that behavioral output is not a perfect reflection of the participant's internal state. The authors address the validity of unconscious WM first by replicating the behavioral phenomenon with some additional manipulations (distractor, load, delay duration). In all cases, they find evidence of 'blindsight'; however, although these manipulations are interesting, it is difficult to see how it addresses the key question – is this just error in visibility reporting? Rather, they turn to the MEG data (which seems sensible). As I see it, the key supporting evidence here is that a neural signature that differentiates seen vs unseen WM (fronto-central beta desynchronization) does not differentiate accuracy within unseen trials (i.e., not a handful of seen, but behaviourally mis-classified trials). I don't actually see how Figure 3 relates to this question, and the results from Figure 5 seem more like a difference of degree, not kind (especially Figure 5, and especially because this is not subdivided by accuracy). So, as to this main (central claim), I remain unconvinced.

2) Interpretation 2. The second challenge to unconscious WM is that participants might actually decide right away (i.e., standard blindsight), and maintain the 'decision' rather than the true event (normal WM of an unconscious percept). I find this a more complicated problem. If participants can decide accurately at the end of the trial, then surely they could also do so at the beginning. If conscious WM were indeed more robust, then it would seem a sensible strategy. Unless, for some reason, they require the response probe to make the blindsight decision? But this is not tested. Rather, the authors suggest that because there is no overlap in the neural markers for conscious (seen vs unseen) and unconscious WM (unseen correct vs unseen error), then they are not likely remembering the blindsight decision. But it is possible that memory for the event rather than the decision have different neural markers. Lower panel of Figure 5 address this somewhat, but the data look pretty noisy, mainly showing that distractor location influences guessing (behavioral data on the link between distractor location and guessing is not show). What would count here as evidence for early decision? The bump around 750ms in middle and right panels would seem suggestive to those inclined, and certainly, it does not look like a very convincing null effect. Especially considering the methodological details listed below.

3) Interpretation 3. Even if there is a prolonged version of blindsight, what is the rationale for calling it WM? Can it be manipulated, for example? Or is it just an extended version of priming?

4) Methodology 1. Statistical power. This study really seems underpowered, especially as some of the main claims are based on null effects. There are only thirteen subjects in the MEG experiment, and only 100 and 200 trials in the perception and WM tasks respectively (small n studies typically compensate with high within-participant power). Bayesian statistics might help convince the reader that the key null effects are meaningful. Also, it would be worth clarifying exactly how many trials are going into these analyses. For example, the behavioral mixture modelling typically requires ~80 trials per subcondition.

5) Methodology 2. Choice of analyses. Firstly, it is not clear what the decoding analysis adds (Figure 3). Something more analogous to the contrasts in Figure 4 would seem more helpful. For example, can a classifier trained on seen vs unseen also classify unseen correct vs unseen incorrect (i.e., accuracy is based on a subset of misclassified visibility ratings). Regarding the location correlation, why not a) decoding (angle, or x,y coordinates) on whole brain activity, b) why not separate correct/incorrect for unseen (assuming unseen correct might actually look like seen trials, statistical power permitting).

6) Methodology 3. The computational model is interesting, but obviously cannot say much about consciousness. I am not qualified to assess the details of the model, but I am assuming it provides a mechanistic account for how some items in WM might be associated with an activity trace, and others not. This is interesting and important, but really only addresses the question of 'activity-silent' WM. Moreover, I would hesitate to call this 'evidence', but rather a proposed model (or even hypothesis).

[Editors' note: further revisions were requested prior to acceptance, as described below.]

Thank you for resubmitting your work entitled "A theory of working memory without consciousness or sustained activity" for further consideration at *eLife*. Your revised article has been favorably evaluated by Tania Pasternak (Reviewing Editor), Sabine Kastner (Senior editor), and two reviewers.

The manuscript has been improved but there are some remaining issues that need to be addressed before acceptance, as outlined below by reviewer 2. Also, include a Table with trial numbers, suggested by reviewer 1.

Reviewer #1:

I was relatively happy with the original version of the manuscript and I believe that it has improved following the revisions. The authors have done a good job in addressing the points raised and should be commended by it. On a minor point, the paper would benefit from a Table containing the mean number of trials on each of the awareness ratings alongside the std.

Reviewer #2:

I appreciate that the authors have worked hard to address my initial concerns, but I am left with a few remaining issues.

1) Interpretation 1. The key claim, as I understand, is that unseen correct cannot be accounted for by a subset of seen correct slipping in (visibility rating error). The main data to support this qualitative difference is the spectral profile (greater beta synchrony for seen vs unseen, not unseen correct vs incorrect; great alpha for unseen correct vs incorrect). However, in response to my original query, the authors also show significant cross-generalisation between the neural signature for seen vs unseen (perception) and unseen correct vs incorrect in the ERF (if I understand this correctly). They subsequently point out that there is no robust discrimination within unseen working memory (i.e., correct vs incorrect), but I do not see that failure to discriminate within the unseen category provides any useful counter evidence at all. Clearly, there is something in the signal that can in principle separate correct and incorrect within the unseen trial (allowing for cross-generalisation); and moreover, that something significantly overlaps with the same something that separates seen and unseen in the perception. Why can't we call this something 'consciousness'? Doesn't this provide exactly the kind of positive evidence we would expect for the mis-classification hypothesis? Why should we weight the difference in beta synchrony over the similarity in ERF topographies? I appreciate that a full dissociation is probably an unrealistically high bar to set – in theory, seen vs unseen need not be neurally orthogonal to unseen correct vs incorrect (e.g., for the reasons given in the rebuttal), but I do not see any principled way to adjudicate the stated hypothesis without requiring a complete dissociation. These results also raise the important question of whether there is an SNR issue (related to my original point about statistical power, see below for further on that). Maybe the decoding analysis is simply a more powerful test of whether there is some 'consciousness' in the unseen correct vs incorrect dimension. It is surprising that the same decoding approach was not used for the beta power. What should we conclude if a seen vs unseen decoder trained on betapower can also significantly discriminate unseen correct vs incorrect? This seems like a very important issue still hanging over the paper. At minimum, I would expect the authors to include the new decoding analyses in the main manuscript, and preferably performed on beta power as well as broadband ERF. If there is a shared discriminative axis between seen vs unseen, and unseen correct vs incorrect, the authors need to clarify exactly how this supports (or otherwise) their central claim. I would also expect the authors to tone-down the claim that the current null effect strictly rules out the misclassification hypothesis. This very strong claim, even if supported by a robust null effect, assumes a strong reverse inference (that beta desynch = consciousness). While this may be true, it is not sufficiently self-evident for a water-tight reverse inference.

2) Interpretation 2. I appreciate the authors' effort to clarify this point, and inclusion of Bayes Factors. Although I am mostly satisfied with their argument, I am also concerned that there is little evidence for decoding conscious memory either, so perhaps it is just not possible to decode memory for persistent ERFs. Conscious or otherwise. That is why I asked about using a more powerful decoding approach (e.g., svm, which could also use the x,y structure of the stimuli quite easily, e.g., svr, which would also handle any non-uniformity around the circle). The authors could also train a classifier on the strongest case (seen correct, and test for generalisation to unseen correct). For a convincing null effect, the reader should feel that a positive effect has been given the best possible chance. Already, the statistical power is really not well suited for this kind of decoding, so at the very least, a state-of-the-art classification approach should be used. Finally, I find the follow-up argument for the circular correlation (channel-wise localisation) unconvincing. Surely, in the context of the current research question, it would be far more important to sacrifice spatial information for optimal classification. I strongly recommend using a proper multivariate classification approach to better test the time course of target (and/or response) location information during unseen correct trials (potentially trained on seen correct).

3) Interpretation 3. I generally agree with the authors' conceptualisation, and appreciate their effort to better clarify the terminology.

4) Methodology 1. I find the arguments in the rebuttal unsatisfying. Evidence of some positive effects does not provide evidence for sufficient power to support null effects (which are central to the key claim of this paper). Bayes Factors help, but are only provided for the decoding analysis (which does not directly speak to the question of consciousness or not in the same way as Figure 3). If the new and unpublished data can help, then they should be included in this paper (not just in a reply to reviewer). I am particularly concerned that one of the central null effects (lack of location decoding during the unseen correct trials) is based on thirteen subjects, and fifty seven unseen trials (further break down of accuracy is not provided). This might be just about OK for strong TFR effects, but not obviously sufficient for subtle decoding effects. At a minimum, the authors need to be clearer about this limitation, and correspondingly more circumspect in the relevant interpretations/conclusions.

5) Methodology 3. Finally, I reiterate my statement that I am not able to critically evaluate the modelling.

---

## [Author Response]

*Essential revisions:*

*1) Please clarify how a disassociated pattern of MEG responses can account for the issue related to errors in visibility reporting (i.e., trials misclassified as unseen).*

We thank the Reviewer for this question as it is a critical point in our argument. In what follows, we will lay out our rationale in more detail and describe the revisions made to the original manuscript.

Previous evidence for non-conscious working memory primarily consists in a behavioral finding: the existence of a long-lasting blindsight effect (i.e., above-chance objective performance with subjectively invisible stimuli after a delay of several seconds). While it is possible that this blindsight genuinely reflects the non-conscious maintenance of information, it is also conceivable that it is an artifact. Subjects may, for example, have mistakenly miscategorized some of the seen trials as unseen. If that were the case, one should be able to identify such miscategorizations based on their neural signature: They should display similar (if not, the same) characteristics of conscious processing as seen trials. If, on the other hand, no such signatures were found on the unseen correct trials, the latter do not just reflect a subset of the seen trials and erroneous visibility reports alone cannot account for the observed blindsight effect. To rule out the miscategorization hypothesis, one therefore needs to show that seen trials are characterized by a neural signature that is absent on the unseen correct trials.

Our time-frequency analysis accomplishes this goal: When compared to target-absent trials, seen trials were characterized by a pronounced decrease in alpha (8–12 Hz) and beta (13–30 Hz) power over fronto-central sensors that was sustained throughout the delay period in the working memory task. A very similar desynchronization was observed when contrasting seen with unseen trials, while no comparable differences emerged between the unseen and target-absent trials (Figure 4 and Figure 4—figure supplement 1). As such, this desynchronization in the alpha/beta band constitutes the signature of conscious processing we set out to identify: It occurs independently of general task context (i.e., perception vs. working memory) and exact inputs (i.e., target location), is specific to seen trials, and prolonged in the working memory as compared to the perception-only control condition.

Importantly, no comparable power decrease was evident on the unseen correct trials. In fact, both unseen correct and unseen incorrect trials resembled each other highly and were indistinguishable from the target-absent control condition (Figure 4 and Figure 4—figure supplement 1). As explained above, the blindsight we observed therefore cannot have resulted from a miscategorization of visibility responses: The clear divergence between seen and unseen correct trials and the absence of any appreciable difference between unseen correct and unseen incorrect/target-absent trials together demonstrate that unseen correct trials were not just mistakenly mislabeled. Please note that we confirmed that the same pattern of results emerged either when a) randomly subsampling the seen trials to have the same number of trials as the average unseen correct condition, or b) restricting our definition of accuracy to within +/- 1 positions of the target (rather than +/- 2).

Although seen and unseen trials were clearly distinct at the brain level, the direct comparison of unseen correct and unseen incorrect trials revealed that these trials were not strictly identical. We found a significant increase in alpha power on the unseen correct compared to the unseen incorrect trials over occipital channels between ~1500 and 1900 ms. Such a difference is reasonable, given that unseen correct and unseen incorrect trials differed in their performance (i.e., the subject identified the correct target location only on the former). Although our experimental design does not allow us to directly test the link between alpha power and task performance, we speculate that, purely by chance, interference from the distractor was reduced on some of the unseen trials compared to others, thus effectively shielding the representation of the target stimulus from being overwritten. Subjects therefore were more likely to later on respond correctly on these than on the other unseen trials.

In order to clarify the distinction between our rationale for rejecting the miscategorization hypothesis (no signature of conscious processing on the unseen correct trials) and the dissociated pattern of MEG responses, we have revised the relevant sections in the Introduction, Results, and Discussion. In particular, we have now included a “roadmap” paragraph, detailing our general logic, at the end of the Introduction; have laid out our hypotheses more clearly in the Results section; have included a supplement to Figure 4 in order to better highlight the signature of conscious processing distinguishing seen from unseen/target-absent trials in all tasks; and have reworked the relevant paragraph in the Discussion section.

*2) Address the question raised by reviewer 1 whether cue detection quantified with d' correlates with working memory performance.*

We thank the Reviewer for this suggestion. While quantifications of perceptual sensitivity do not allow one to draw inferences about whether or not conscious awareness was reached (e.g., Soto et al., 2011), it is indeed interesting to examine whether an individual’s sensitivity to the target correlates with his or her objective task performance. After all, perhaps those subjects who were better able to discriminate signal from noise, might also be the ones to show the most blindsight (which, then, could suggest that conscious perception and working memory might not be entirely decoupled).

We computed detection d’ (z(hits) – z(false alarms), where hits = proportion of seen target-present trials and false alarms = proportion of seen target-absent trials) in all experiments and correlated it with objective task performance separately for the seen and unseen trials. The results have been added to the respective paragraphs in the Results section and a supplement has been added to Figure 2 to depict those correlations. Briefly, in both the perception and working memory task in experiment 1, target detection d’ exceeded chance (perception: d’ = 1.5 +/- 0.9, t(16) = 7.1, p<0.001; working memory: d’ = 1.5 +/- 0.7, t(16) = 8.9, p<0.001). Such above-chance d’ was, of course, part of our design, since we aimed to obtain a mixture of seen and unseen stimuli for a fixed stimulus. Importantly, on unseen trials, participants’ sensitivity to the target did not correlate with any of our measures of task performance (accuracy: r = 0.320, p=0.210; rate of correct responding: r = 0.342, p=0.179; precision: r = 0.114, p=0.710). Experiment 2 replicated these results. Detection d’ was significantly greater than chance (1.7 +/- 0.8, t(20) = 10.2, p<0.001), yet uncorrelated with accuracy (r = 0.366, p=0.124), the rate of correct responding (r = 0.224, p=0.357), or precision (r = -0.410, p=0.115) on unseen trials. In both experiments 1 and 2, however, sensitivity to the target correlated positively with accuracy and the rate of correct responding on seen trials (all rs > 0.443, all ps<0.051). Objective task performance on the unseen trials was thus dissociated from perceptual awareness.

*3) Please discuss if the "activity-silent" code represents a more general property or is specific to non-conscious working memory as well as discuss what types of WM processes are more likely dependent on consciousness.*

We thank the Reviewer for having brought up this question as it allows us to clarify our stance. We indeed believe that the activity-silent code is not specific to non-conscious working memory and also contributes to conscious working memory. While initially encoded and maintained through slowly decaying neuronal firing, the representation of seen target locations is also not continuously sustained, but waxes and wanes throughout the retention period (Figure 5). As in our simulations (Figure 7), there thus exist activity-silent phases interspersed with periods of spontaneous reactivations (potentially, as discussed in more detail in the manuscript, reflecting conscious rehearsal). We have fully reworked all relevant parts of the Results and Discussion section in order to emphasize this common mechanism underlying conscious and non-conscious working memory.

The Reviewer also raises another intriguing possibility: Perhaps activity-silent representations reflect a more general property of memory that goes beyond working memory. We believe this to be a very relevant and reasonable comment indeed and have since included a discussion of this point.

As concerns the second part of the Reviewer’s comment, while we indeed hope to demonstrate that non-conscious working memory is genuine and that, as such, a decoupling of working memory and conscious perception is not just an artifact, we do not want to claim that there is no benefit of conscious awareness to working memory. Subjects, for example, clearly perform much better on the seen than on the unseen trials. In addition, in the present experiments, we specifically chose to focus on non-conscious maintenance of information, because this lies at the heart of most conceptualizations of working memory (see also Eriksson et al., 2015), but this does not mean that all working memory processes can necessarily occur non-consciously. It would indeed be of great value for future research to assess whether other aspects of working memory, such as manipulation of information, may also be dissociated from consciousness. We have therefore added a paragraph to the Discussion, addressing this issue and cautiously speculating that, in light of the proposed passive activity-silent mechanism for non-conscious working memory, it is unlikely that processes requiring neural activity (such as transformations of the stored representations, e.g., mental rotation) can proceed in the complete absence of awareness.

*4) Address the problem of the accuracy on trials reported as unseen (i.e trials misclassified as unseen), raised by Rev 2 ("Interpretation 1").*

We believe that this point is closely related to Essential revision 1 and therefore extend our arguments. The Reviewer is correct that one of our objectives consisted in testing the reality of non-conscious working memory by examining the previously reported long-lasting blindsight effect in light of two alternative hypotheses: the miscategorization hypothesis (i.e., subjects erroneously labeled seen trials as unseen) and the conscious maintenance hypothesis (i.e., subjects consciously maintained an early guess). This was, however, embedded in a much larger investigation with several key questions: First, we wanted to examine the robustness of the blindsight with regard to interference from distraction and a conscious working memory load, as this allowed us to further characterize its basic properties. These behavioral results are reported in Figure 1 and Figure 2 We then directly probed the relationship between conscious perception and conscious working memory, specifically focusing on the hypothesis that maintenance in conscious working memory would constitute a prolonged conscious episode. To this end, we employed multivariate pattern analyses and reported the results in Figure 3. The Reviewer is thus right: Figure 3 does not directly relate to the question of the genuineness of non-conscious working memory; Figure 4, Figure 5, and 6 do. We regret that our presentation of our guiding objectives had not been sufficiently clear and have since substantially revised the paper to rectify this issue. Specifically, we have included a “roadmap” for our rationale at the end of the Introduction and have better introduced each main objective (and any specific hypotheses) in the corresponding paragraphs of the Results section. In the Discussion, we have also reworked the summary paragraph and conclusion in order to better tie in our findings with our guiding principles.

Furthermore, we agree with the Reviewer that the key evidence against the miscategorization (and as explained below, the conscious maintenance) hypothesis is that the signature of conscious processing (i.e., a desynchronization in the alpha/beta band) does not differentiate the unseen correct from the unseen incorrect trials (Figure 4). Indeed, as also laid out in response to reviewer 1 (Essential revision 1), if the long-lasting blindsight effect resulted from a subset of seen trials that had erroneously been reported as unseen, the very same signature of conscious processing, comparable in size to seen trials, should also be present on the unseen correct trials (especially so, as, due to our task design, chance is very low and the majority of the unseen correct trials reflect true task performance rather than random guessing). This is, however, not the case. While seen trials in both the perception-only control condition and the working memory task displayed a prominent decrease in alpha/beta power (as compared to target-absent and unseen trials), no such desynchronization was present on the unseen trials – not even when considering them separately by accuracy (Figure 4 and Figure 4—figure supplement 1). Bayesian statistics further supported this conclusion (Figure 4—figure supplement 3): In the beta band, evidence for the alternative hypothesis (i.e., a power decrease/increase relative to baseline) was consistently higher than for the null hypothesis (i.e., no power decrease/increase relative to baseline) for the seen trials. In contrast, for the unseen trials, the null hypothesis was generally favored over the alternative hypothesis. The observed decrease in alpha/beta power is thus specific to seen trials and, importantly, was first identified in the perception-only control condition without any maintenance requirement: It is a general signature of conscious processing. In addition, although these data are not shown, we carefully ensured that the very same signature persisted, even when subsampling the seen trials to the average number of unseen correct trials. Besides a small increase in alpha power around the time of the distractor, no differences emerged between the unseen correct and the unseen incorrect trials. As such, the miscategorization hypothesis cannot account for the observed blindsight effect.

*5) Address a possibility, raised by reviewer 2, that subjects retain the "blindsight" decision to the end of the trial ("Interpretation 2") and that memory for the event and for the decision may have different neural markers.*

Two important points were raised here. The first one concerns one of the main challenges to non-conscious working memory: Perhaps, just as in standard priming experiments, participants guessed the target location right after its presentation and then consciously maintained this above-chance decision (conscious maintenance hypothesis). The present study tackles this problem for the very first time, by relying on two main lines of converging evidence. On one hand, if participants consciously maintained a guess, then there should be a signature of conscious processing on the unseen trials. As explained in more detail above, this was not the case: There was no trace of any decrease in alpha/beta power on the unseen trials even remotely comparable to the one on seen trials (Figure 4 and Figure 4—figure supplement 1; Figure 4—figure supplement 3). Importantly, this desynchronization in the alpha/beta band constitutes a task- and stimulus-independent, general signature of conscious processing (i.e., not just a marker of a specific event). It was first established in our perception-only control condition (without any working memory requirement) and is blind as to specific target and/or response locations. Even if the format of a specific representation changed depending on whether or not it was a memory of the target stimulus or a memory of the decision, the absence of any appreciable decrease in alpha/beta power on the unseen trials suggests that subjects did not just maintain a conscious guess.

Circular-linear correlation analyses further supported this conclusion (Figure 6): While response location could be tracked throughout the majority of the epoch on seen trials (albeit through a slowly decaying and fluctuating code), the representation of response position on the unseen trials did not come online until much later, towards the very end of the epoch. In line with previous research (Bode et al., 2011), this build-up of activity could potentially reflect the non-conscious generation of our subjects’ guess, although future research will be needed to confirm this speculation. At the very least, this absence of decodability suggests that, for the vast majority of our epoch, participants did not maintain a conscious guess. This interpretation was validated by Bayesian statistics ([Supplementary-material SD3-data]), favoring the null hypothesis (i.e., no decodable information) over the alternative hypothesis (i.e., decodable information) for the vast majority of our temporal ROIs on the unseen trials. Even more importantly, combining the current data with the data from an ongoing study on the non-conscious manipulation of information (see our response to Essential revision 7 for further details), we replicated these findings in a larger sample of 29 participants (see Figure 10 in response to Essential revision 7), thus reconfirming the present conclusions with much higher confidence.

The second point raised by the reviewer concerns a potential dissociation of the neural markers for memories of events vs. memories of decisions. While, as explained above, we believe that we identified a general signature of conscious processing and, as such, already addressed this possibility in our manuscript, we nevertheless returned to our circular-linear correlations to further interrogate this issue. Specifically, we carefully examined the topographical patterns related to a subject’s response. Just as was the case for the correlations with target location, focal associations between the response location and the MEG signal were again primarily confined to posterior channels, with more frontal areas being recruited exclusively at the time of the response (Figure 6). No additional regions were identified on the unseen trials and none of these areas showed any appreciable correlation before the presentation of the response screen (Figure 6—figure supplement 1). This suggests that, irrespective of stimulus visibility, common brain networks supported memories for the target stimulus and the ensuing decision and that, in the case of non-conscious working memory, these did not come online until shortly before the response. These findings were again highly reproducible across different experiments (current study and ongoing investigation; Figure 8). We have now modified the respective parts of the Results section and separately present the topographies and time courses of the circular-linear correlations with the response location in Figure 6 and Figure 6—figure supplement 1.

Topographies of the circular-linear correlations with response location are shown for seen (left) and unseen (right) trials across different experiments. The first row corresponds to the present study, the second to an ongoing investigation. Data from the two experiments were combined in the third row. The first three time bins are relative to stimulus onset, the last two relative to response screen onset. R = onset of response screen.

Author response image 1.**DOI:**
http://dx.doi.org/10.7554/eLife.23871.023

*6) Clarify the reasons for referring to extended duration "blindsight" as working memory (reviewer 2, "Interpretation 3").*

We thank the Reviewer for this comment, as it allows us to clarify our rationale (please see also our response to Essential revision 3). It is simply a truism that the short-term maintenance of information lies at the heart of most conceptualizations of working memory (see Eriksson et al., 2015 for a review) and the existing literature on non-conscious working memory (e.g., Soto et al., 2011). “Blindsight” is the generic term given to any situation of above-chance objective performance in the absence of a subjective feeling of consciousness. Given the current state of the evidence, “non-conscious working memory” seems to be a more accurate descriptor of the phenomenon that we study than the generic term of “blindsight.” First, the time scales investigated here and in previous reports of non-conscious working memory (e.g., Soto et al., 2011; Bergström et al., 2015) lie far beyond the ones observed in typical priming experiments, going up to delays of 15 s (Bergström et al., 2014). Second, and perhaps most importantly, the observed blindsight exhibits a lot of the same characteristics as would be expected from conscious working memory, including resistance to distraction and sensitivity to load manipulations. Third, we purposefully designed our experiments to include a large number of stimulus locations and assigned the corresponding response cues randomly on each trial. It is thus unlikely that participants’ location responses were primarily driven by automatic stimulus-response mappings and, instead, required a minimal amount of manipulation.

Following the Reviewer’s comment, we added a paragraph mentioning that there are also additional criteria for working memory, such as the mental manipulation of information, that were not interrogated in the current paper (subsection “Long-lasting blindsight effect reflects genuine non-conscious working memory”). Whether such processes can occur non-consciously is an important empirical question and should be addressed by future research. In the added discussion, we cautiously speculate that, in light of the proposed activity-silent model for non-conscious working memory, it seems unlikely that processes requiring a transformation of a neural code could occur in the complete absence of awareness.

*7) Please address the issue of statistical power raised by reviewer 2 ("Methodology 1"). Specifically, the power of analysis is limited by the small number of error trials.*

We are confident that our study possesses the necessary statistical power to examine the question at stake and believe that it establishes an important methodological and theoretical framework for future research. First, both our sample size and number of trials (200 each in both the perception and working memory task) appear to be well into the acceptable range for MEG experiments of this kind (i.e., a working memory task with long trials). Our stimuli were tailored in a subject-specific manner to obtain a large proportion of seen and unseen trials. After rejection of artifacted trials, there were, on average 73 (71) seen and 61 (57) unseen target-present trials in the perception (working memory) task. Second, and most importantly, throughout the entire manuscript we repeatedly demonstrate that we are able to detect generally well-established and subtle effects specifically on the unseen trials – often with magnitudes similar to the ones observed on seen trials:

1) Behaviorally, a conscious working memory load modulated the precision with which non-conscious (but not conscious) information could be stored (Figure 2).

2) When comparing the event-related fields (ERFs) of seen to unseen trials, we detected a well-established signature of conscious perception (i.e., an ignition of activity specific to the seen trials; Figure 3).

3) As part of our time-frequency analyses, we show a significant increase in alpha power on the unseen correct compared to the unseen incorrect trials (Figure 4).

4) During our early time window (100–300 ms), we are consistently able to decode target location on the unseen (as well as the unseen correct) trials, equivalently to seen trials (as assessed in a paired Wilcoxon signed-rank test; Figure 5).

5) Similarly, we can track response location with equal magnitude on the seen and unseen trials at the time of the response and, in left temporo-occipital and frontal channels, with similar size on the unseen correct and unseen incorrect trials (Figure 6). Even more intriguingly, towards the end of the epoch (right before the response), response location ramps up on the unseen trials in all sensors (but most pronouncedly in the frontal ones), demonstrating that, even during the delay period, we can capture subtle events on the unseen trials.

We have now highlighted these instances throughout the manuscript.

In addition, we have made further efforts to demonstrate that our key effects are meaningful. We have now added Bayesian statistics, which generally supported our theoretical interpretations, showing that a) there is essentially no evidence for any change in beta power (relative to baseline) on the unseen trials until the very end of the epoch (Figure 4—figure supplement 3) and that b), similarly, on the unseen trials, the absence of any target- and response-related activity during the delay period is overall more likely than their presence ([Supplementary-material SD3-data]).

We recently replicated these findings as part of a new study on the same topic, which used MEG to examine the possibility of non-conscious manipulation of information. The control condition of this new experiment is strictly equivalent to the current study: Subjects were asked to maintain a target square over a 3s-delay period and to report a location later on. We combined the data from this control with the current experiment (29 participants total) and reran the circular-linear correlation analyses. As shown in Figure 9 and Figure 10 below, this confirmed the absence of any delay-activity on the unseen trials.

Author response image 2.(**A**) Time courses of circular-linear correlations between MEG signals and target location combined across data from 29 participants for seen (red) and unseen (blue) trials. In the replication study, a visual cue was presented 1.75 s after the presentation of the target stimulus, indicating specific response modalities. (**B**) Time courses of circular-linear correlations between MEG signals and target location combined across data from 29 participants for unseen correct (dark blue) and unseen incorrect (light blue) trials.**DOI:**
http://dx.doi.org/10.7554/eLife.23871.024

Author response image 3.(**A**) Time courses for circular-linear correlations between MEG signals and response location combined across data from 29 participants for seen (red) and unseen (blue) trials. In the replication study, a visual cue was presented 1.75 s after the presentation of the target stimulus, indicating specific response modalities. (**B**) Time courses for circular-linear correlations between MEG signals and response location combined across data from 29 participants for unseen correct (dark blue) and unseen incorrect (light blue) trials.**DOI:**
http://dx.doi.org/10.7554/eLife.23871.025

*8) For decoding, use a classifier trained on "seen" vs "unseen" trials to distinguish between "unseen correct "vs "unseen incorrect" (Reviewer 2, "Methodology 2").*

We thank the Reviewer for this suggestion and describe the results of this analysis here. In order to ensure a maximum of samples while strictly separating the train and test sets, we trained a classifier to distinguish seen from unseen trials in the perception task, and then applied it either to the same category of trials in the working memory task (visibility decoder) or to the unseen correct and incorrect trials in the working memory task (accuracy decoder). The temporal generalization matrix for the seen/unseen classification presented as a thick diagonal, indicating that brain responses consisted in a succession of distinct patterns of activity that partially overlapped in time. Diagonal classification performance exceeded chance-level between ~148 and 1048 ms (*FDR*-corrected). When applied to the unseen correct and incorrect trials, the same classifiers generalized only weakly: Using a lenient, uncorrected threshold, diagonal decoding rose above chance between ~372 and 724 ms. Importantly, these findings are compatible with two different interpretations. On one hand, they might indicate that unseen correct trials represent seen trials that had been mislabeled. Alternatively, the model of non-conscious working memory we propose suggests that a stimulus that fails to cross the threshold for sustained activity and subjective visibility (“failed ignition”), is still expected to induce enough activity in high-level cortical circuits to trigger short-term synaptic changes. As such, unseen correct trials should share some of the processes that are found on seen trials and a decoder should therefore generalize to some extent from seen to unseen correct (as observed in Figure 11).

Author response image 4.(**A**) Temporal generalization matrices for classifiers trained to distinguish seen from unseen trials in the perception task and applied to the working memory task. Horizontal, dotted line denotes mean reaction time for the visibility response in the perception task. Inset represents the diagonal of the matrix (black), i.e. classifiers were trained and tested on the same time sample. Thick line indicates above-chance decoding performance (one-tailed Wilcoxon signed-rank test, uncorrected). Shaded area denotes standard error of the mean (SEM) across subjects. For display purposes only, data were smoothed with a moving average of eight samples. (**B**) Same as in (**A**), except that classifiers trained to distinguish seen from unseen in the perception task were applied to unseen correct and unseen incorrect trials in the working memory task.**DOI:**
http://dx.doi.org/10.7554/eLife.23871.026

In order to separate these two hypotheses, we trained and tested a classifier separately on seen/unseen correct and seen/unseen incorrect trials in the working memory task. If the majority of our unseen correct trials were in fact seen trials mislabeled as unseen, then brain responses should be highly similar in the two conditions and classifiers should perform at chance. As can be seen in Figure 12, this was not the case. Instead, classifiers trained to separate seen and unseen correct trials performed significantly above chance from ~332 to 992ms, thus revealing a clear divergence in the brain responses elicited by each condition. In fact, the diagonal-shape pattern revealed by temporal generalization was similar to the ones observed when classifiers were trained to separate seen from either all unseen trials (Figure 12) or from unseen incorrect trials (Figure 12). Classifiers trained to distinguish unseen correct from unseen incorrect trials did not show any clear difference between these conditions (Figure 12). These results demonstrate that brain responses in unseen correct trials were clearly different from seen trials but not from unseen incorrect trials. As such, these findings converge with the results obtained from the time-frequency analysis and reject the hypothesis of a miscategorization of unseen correct trials. Unseen correct trials rather reflect genuine blindsight, in which subjects correctly reported the location of the target stimulus while being unaware of it.

Author response image 5.Temporal generalization matrices for classifiers trained to distinguish seen from unseen trials (**A**), seen from unseen correct (**B**), seen from unseen incorrect (**C**), and unseen correct from unseen incorrect (**D**) in the working memory task. Insets represent the diagonal of the matrix (black), i.e. classifiers were trained and tested on the same time sample, and P3b (red) slices, i.e., classifiers trained between 0.3 and 0.6 s were averaged. Thick line indicates above-chance decoding performance (one-tailed Wilcoxon signed-rank test, uncorrected). Shaded area denotes standard error of the mean (SEM) across subjects. For display purposes only, data were smoothed with a moving average of eight samples. Note that a class weight was applied to counter any class imbalances.**DOI:**
http://dx.doi.org/10.7554/eLife.23871.027

[Editors' note: further revisions were requested prior to acceptance, as described below.]

*Reviewer #1:*

*I was relatively happy with the original version of the manuscript and I believe that it has improved following the revisions. The authors have done a good job in addressing the points raised and should be commended by it. On a minor point, the paper would benefit from a Table containing the mean number of trials on each of the awareness ratings alongside the std.*

We thank the Reviewer for his/her positive evaluation of our work and are happy to hear that our revisions were satisfactory. The requested table has been added to our manuscript ([Supplementary-material SD4-data]).

*Reviewer #2:*

*I appreciate that the authors have worked hard to address my initial concerns, but I am left with a few remaining issues.*

*1) Interpretation 1. The key claim, as I understand, is that unseen correct cannot be accounted for by a subset of seen correct slipping in (visibility rating error). The main data to support this qualitative difference is the spectral profile (greater beta synchrony for seen vs unseen, not unseen correct vs incorrect; great alpha for unseen correct vs incorrect). However, in response to my original query, the authors also show significant cross-generalisation between the neural signature for seen vs unseen (perception) and unseen correct vs incorrect in the ERF (if I understand this correctly). They subsequently point out that there is no robust discrimination within unseen working memory (i.e., correct vs incorrect), but I do not see that failure to discriminate within the unseen category provides any useful counter evidence at all. Clearly, there is something in the signal that can in principle separate correct and incorrect within the unseen trial (allowing for cross-generalisation); and moreover, that something significantly overlaps with the same something that separates seen and unseen in the perception. Why can't we call this something 'consciousness'? Doesn't this provide exactly the kind of positive evidence we would expect for the mis-classification hypothesis? Why should we weight the difference in beta synchrony over the similarity in ERF topographies? I appreciate that a full dissociation is probably an unrealistically high bar to set – in theory, seen vs unseen need not be neurally orthogonal to unseen correct vs incorrect (e.g., for the reasons given in the rebuttal), but I do not see any principled way to adjudicate the stated hypothesis without requiring a complete dissociation. These results also raise the important question of whether there is an SNR issue (related to my original point about statistical power, see below for further on that). Maybe the decoding analysis is simply a more powerful test of whether there is some 'consciousness' in the unseen correct vs incorrect dimension. It is surprising that the same decoding approach was not used for the beta power. What should we conclude if a seen vs unseen decoding trained on beta power can also significantly discriminate unseen correct vs incorrect? This seems like a very important issue still hanging over the paper. At minimum, I would expect the authors to include the new decoding analyses in the main manuscript, and preferably performed on beta power as well as broadband ERF. If there is a shared discriminative axis between seen vs unseen, and unseen correct vs incorrect, the authors need to clarify exactly how this supports (or otherwise) their central claim. I would also expect the authors to tone-down the claim that the current null effect strictly rules out the misclassification hypothesis. This very strong claim, even if supported by a robust null effect, assumes a strong reverse inference (that beta desynch = consciousness). While this may be true, it is not sufficiently self-evident for a water-tight reverse inference.*

We thank the Reviewer for this comment and have since implemented all of the suggestions. In what follows, we first clarify our reasoning, describe the main findings of the new decoding analyses, and then highlight the changes we implemented in the manuscript.

Adjudicating between the miscategorization hypothesis and genuine non-conscious working memory involves determining whether the unseen correct trials are more similar to the seen trials or to the unseen incorrect trials. In response to the Reviewer’s previous comments, we presented decoding analyses aimed at assessing the overlap in neural patterns between the seen and unseen correct trials (see our response to Essential revision 8). We trained a classifier to separate seen from unseen trials in the perception task and then applied it either to the seen and unseen trials in the working memory task (decoding visibility) or to the unseen correct and unseen incorrect trials in the working memory task (decoding accuracy). Decoding visibility presented as a thick diagonal, with above-chance discrimination between ~148 and 1048 ms (Figure 4—figure supplement 2, left): Classification performance first rose above chance at ~148 ms (AUC = 0.54 +/- 0.01, p_FDR_=0.023), peaked at ~640ms (AUC = 0.58 +/- 0.02, p_FDR_=0.001), and then decayed rapidly by ~1 s. In contrast, the generalization to the unseen correct/incorrect trials was barely significant, did not reveal any clear pattern, and the time course of diagonal decoding was highly dissimilar from the one observed above (Figure 4—figure supplement 2, right): It first sharply peaked at ~180 ms (AUC = 0.55 +/- 0.01, p_uncorrected_=0.037), dropped to chance-level, and then exceeded chance between ~372 and 724 ms with a peak at 444 ms (AUC = 0.57 +/- 0.02, p_uncorrected_=0.007). Much unlike any of the previous decoders involving the perception task, long after the visibility response, it rose a third time between ~1.44 and 1.74 s, peaking with similar magnitude as before at ~1.58 s (AUC = 0.57 +/- 0.02, p_uncorrected_=0.010; P3b and last time window: all ps<0.023). Although the level of noise evident in the accuracy decoder thus precludes any definitive conclusion, the visibility and accuracy decoders had little in common, rendering it unlikely for the unseen correct trials to have simply been mislabeled.

Importantly, even if this weak effect were genuine, on its own, this analysis is not well suited to distinguish among the two competing hypotheses, as it is only capable of assessing whether the two categories of trials might share common processes (without specifying the exact nature). In fact, a weak and partial generalization from seen/unseen to unseen correct/incorrect is not only compatible with, but actually expected by our proposed model of activity-silent non-conscious working memory, in which the unseen correct trials, while failing to cross the threshold for subjective visibility, behave more similar to the seen than the unseen incorrect trials in that they still induce enough activity in higher-level cortical circuits to modify synaptic weights. Similarly, it is conceivable that a partial generalization results simply from the fact that both the seen as well as the unseen correct trials share a common potential for correct responding.

The critical test thus consists in directly comparing the seen to the unseen correct trials, and the unseen correct to the unseen incorrect trials (see also our previous response to Essential revision 8). According to the miscategorization hypothesis, seen and unseen correct trials should, in fact, belong to the same category, whereas unseen correct and incorrect trials should not. A classifier trained to discriminate between seen and unseen correct trials should thus perform at chance and a classifier trained to distinguish the unseen correct from their incorrect counterpart should resemble the standard seen/unseen as well as a seen/unseen incorrect decoder. We observed the exact opposite pattern: The unseen correct/incorrect classification was at chance, while decoding of seen/unseen correct trials revealed a pattern of brain responses highly similar to the one observed for seen/unseen incorrect. Because there were about twice as many seen trials as unseen correct/incorrect trials, we repeated the same analysis, this time randomly subsampling the seen trials (and, in the case of the seen/unseen classification, the unseen trials as well) to the average number of unseen correct trials. The results remained virtually unchanged: All generalization matrices involving seen trials presented as thick diagonals, with above-chance diagonal-decoding at least between ~320 and 800 ms (in the case of the seen/unseen correct decoder). There were no apparent differences between decoding with the full and subsampled datasets. The unseen correct and incorrect trials, however, could not be discriminated at all. As such, these results do not support the miscategorization hypothesis. They are, however, compatible with the hypothesis of genuine non-conscious working memory.

In a second step, we also implemented the Reviewer’s suggestion and extended the decoding approach to the frequency domain. We first followed the same logic as for the univariate analyses, training classifiers on the average power (in dB, relative to baseline) in the alpha (8–12 Hz) or beta (13–30 Hz) band to distinguish seen from unseen trials separately in the perception and working memory task and applying it to the same visibility categories either in the same or the other task (Figure 13). This analysis largely confirmed our univariate time-frequency results: Temporal generalization matrices for alpha and beta power were characterized by thick diagonals in the perception task, with diagonal decoding exceeding chance between ~440 and 1220 ms in the alpha (Figure 13A; P3b [300–600 ms] time window and first part of the delay [0.6–1.5 s]: ps<0.027) and between ~100 and 180 ms, ~600 and 800 ms, as well as between ~1.26 and 1.4 s in the beta band (Figure 13; early [100–300m s] interval and first part of the delay period: ps <0.027). A similar, yet more sustained, pattern was also observed in the working memory task, although temporal generalization matrices for both bands were slightly more square-shaped than in the perception task and diagonal decoding persisted until the end of the epoch (alpha: last two time windows, ps<0.019; beta: 0.6–1.5 s, p=0.004; 1.5–2.2 s, p=0.066). Importantly, generalization from one task to the other resulted in a similar pattern of decodability in the alpha band. When trained in the perception task and tested in the working memory task, diagonal decoding emerged around 180 ms and lasted until 1.18 s (first three time windows: ps <0.016). Probably owing to the slightly different time courses of beta desynchronization in the perception and working memory task (Figure 4—figure supplement 1), decoding in the beta band was less strong and, if anything, tended a bit more towards the off-diagonal, such that diagonal decoding itself occurred mainly between ~320 and 540 ms as well as between 1.24 and 1.78 s, yet reached statistical significance in all four time bins (all ps<=0.05). The reverse generalization, from the working memory to the perception task, revealed a similar set of findings, with diagonal decoding in the alpha band occurring between ~100 ms and 1.44 s (first three time windows: ps<0.037) and in the beta band between ~100 and 540 ms and again between ~940 ms and 1.8 s (all time windows: ps<0.013). Taken together, these multivariate analyses are in-line with our previous univariate results, suggesting that power in the alpha/beta band may serve as a signature of conscious processing.

Last, in analogy to the ERF decoding analysis carried out in response to the Reviewer’s previous comments (Figure 4—figure supplement 2), we also applied the aforementioned visibility decoders trained in the perception task to the unseen correct and incorrect trials in the working memory task separately for the alpha (Figure 4—figure supplement 2) and beta (Figure 4—figure supplement 2) band. There was no clear evidence for any generalization in either band. In light of the somewhat weaker visibility generalization in the beta band, the absence of decodability in this band might be difficult to interpret unequivocally with the current dataset. However, the fact that, despite a good initial visibility classifier, no generalization to the unseen correct trials occurred in the alpha band either argues against a simple miscategorization of the unseen correct trials.

Taken together, we presented a series of analyses aimed at evaluating the miscategorization hypothesis and, overall, found a clear distinction in the brain responses of seen and unseen (correct) trials. We were unable to classify unseen correct and unseen incorrect trials in the ERF; the univariate time-frequency analyses show a clear dissociation between the seen and unseen correct trials; there was no generalization between the seen and unseen correct trials in either the alpha or the beta band. The bulk of the evidence thus repeatedly converges towards a rejection of the miscategorization hypothesis.

We nevertheless agree with the Reviewer that, due to the nature of the research question at hand, it is difficult – if not impossible – to unequivocally rule out the miscategorization hypothesis in a single study. We believe that our paper represents an important first step in this direction and offers an interesting theoretical basis, but, ultimately, future research with a higher signal-to-noise ratio is needed to replicate these findings. In the revised version of this manuscript, we have therefore toned down our claim whenever appropriate and have added a final paragraph to the Discussion, pointing to the limitations of our study as well as the ensuing perspectives for future research. We also added the results of the generalization from visibility to accuracy decoding to the main manuscript ( Figure 4—figure supplement 2).

Author response image 6.(**A**) Temporal generalization matrices for decoding of visibility category with relative, average alpha (8–12 Hz) power as a function of training and testing task. In each panel, a classifier was trained at every time sample (y-axis) and tested on all other time points (x-axis). The diagonal gray line demarks classifiers trained and tested on the same time sample. Please note the event markers in any panel involving the perception task: Mean reaction time (target-present trials) for the visibility response is indicated as vertical and/or horizontal, dotted lines. Any classifier beyond this point only reflects post-visibility processes. Time courses of diagonal decoding and of classifiers averaged over the P3b time window (300–600 ms) and over the working memory maintenance period (0.8–2.5 s) are shown as black, red, and blue insets. Thick lines indicate significant, above-chance decoding of visibility (Wilcoxon signed-rank test across subjects, uncorrected, two-tailed except for diagonal). For display purposes, data were smoothed using a moving average with a window of one sample. AUC = area under the curve. (**B**) Same as in (**A**) but for average beta (13–30 Hz) power.**DOI:**
http://dx.doi.org/10.7554/eLife.23871.028

*2) Interpretation 2. I appreciate the authors' effort to clarify this point, and inclusion of Bayes Factors. Although I am mostly satisfied with their argument, I am also concerned that there is little evidence for decoding conscious memory either, so perhaps it is just not possible to decode memory for persistent ERFs. Conscious or otherwise. That is why I asked about using a more powerful decoding approach (e.g., svm, which could also use the x,y structure of the stimuli quite easily, e.g., svr, which would also handle any non-uniformity around the circle). The authors could also train a classifier on the strongest case (seen correct, and test for generalisation to unseen correct). For a convincing null effect, the reader should feel that a positive effect has been given the best possible chance. Already, the statistical power is really not well suited for this kind of decoding, so at the very least, a state-of-the-art classification approach should be used. Finally, I find the follow-up argument for the circular correlation (channel-wise localisation) unconvincing. Surely, in the context of the current research question, it would be far more important to sacrifice spatial information for optimal classification. I strongly recommend using a proper multivariate classification approach to better test the time course of target (and/or response) location information during unseen correct trials (potentially trained on seen correct).*

We are glad to see that the Reviewer is mostly satisfied with our response and hope that we will be able to successfully address the remaining points. Indeed, while statistically significant at least throughout the first part of the delay period, decodability of seen targets was also not continuously sustained, but waxed and waned throughout the delay. While we interpret this periodically disappearing target-related activity not as an artifact of our analyses, but as a genuine neural mechanism supporting maintenance in conscious working memory, we fully agree with the Reviewer that a positive effect needs to have been given the best possible chance of being detected. We therefore implemented the Reviewer’s suggestion in order to test whether target and response location could be decoded using a multivariate regression approach. These results are depicted in Figure 5—figure supplement 2, Figure 5—figure supplement 3, and Figure 6—figure supplement 2 of the manuscript.

Following King et al. (2016), for each participant and each experimental condition, we trained two separate support vector regression models to predict the sine and cosine of the target/response location in question and then combined the resulting predictions to estimate the target angle. As can be seen in Figure 5—figure supplement 2 and Figure 6—figure supplement 2, the decoding time courses, although noisier, were very similar to the ones obtained with the circular-linear correlations and led us to the same conclusions: Seen targets/response locations were maintained at least throughout the first part of the delay period (P3b time window and first part of the delay: ps<0.05) via a slowly decaying and periodically resurfacing code, whereas no such pattern was evident on the unseen trials. When applying our best classifier (seen correct trials), separately to the unseen correct and incorrect trials, no generalization was observed for either target (Figure 5—figure supplement 3) or response location (Figure 5—figure supplement 3). The decoding approach thus confirmed the results from our circular-linear correlation analyses, albeit with more noise (cf., stability of standard error for the circular-linear correlations across different conditions vs. increases in standard error for the multivariate approach across different conditions).

Although, at first glance, the more stable performance of the circular-linear correlations might be surprising, it is understandable given the experimental paradigm: There are twenty target locations and, on average, only ~60 seen/unseen trials, so using a proper cross-validation scheme with a strict separation of the train and test sets makes it virtually impossible to build an adequate classifier. For example, even when combining across seen and unseen trials in the working memory task, a minimal two-fold cross-validation procedure would only include ~3 instances of each target/response location in each of the folds. If the initial classifier is poor because of limited data to train with, so will be the resulting predictions. This is obviously a limitation of the present investigation (see also below) that should be addressed by future research. Circular-linear correlations, on the other hand, combine information from all available trials and are thus more powerful in the current context. In addition, they also result in easily interpretable topographies, which, in the context of a spatial paradigm, is important information to consider. As such, circular-linear correlations appear to be the most appropriate measure in the context of the current investigation.

*3) Interpretation 3. I generally agree with the authors' conceptualisation, and appreciate their effort to better clarify the terminology.*

We are happy to hear that our efforts were successful.

*4) Methodology 1. I find the arguments in the rebuttal unsatisfying. Evidence of some positive effects does not provide evidence for sufficient power to support null effects (which are central to the key claim of this paper). Bayes Factors help, but are only provided for the decoding analysis (which does not directly speak to the question of consciousness or not in the same way as Figure 3). If the new and unpublished data can help, then they should be included in this paper (not just as a reply to reviewer). I am particularly concerned that one of the central null effects (lack of location decoding during the unseen correct trials) is based on thirteen subjects, and fifty seven unseen trials (further break down of accuracy is not provided). This might be just about OK for strong TFR effects, but not obviously sufficient for subtle decoding effects. At a minimum, the authors need to be clearer about this limitation, and correspondingly more circumspect in the relevant interpretations/conclusions.*

Bayes Factors had in fact been reported for all key analyses, including the time-frequency analysis (Figure 4—figure supplement 2) and the circular-linear correlations (Results section and [Supplementary-material SD3-data]). In addition, we had included data from our ongoing investigation in the – publicly available – response letter in order to draw attention to the fact that, even when doubling our sample size, the reported null effects still persist. While some conditions in this new experiment are sufficiently similar to the current manuscript to allow for averaging across all trials, the underlying experimental questions are different and will therefore be reported in the future. Nevertheless, the results of the condition shared with the Reviewer serve as a first step towards demonstrating the replicability of these findings.

While the lack of location decoding on the unseen correct trials is certainly of empirical and theoretical interest (e.g., the proposed model would actually predict for unseen correct target locations to fall somewhere in-between the seen and unseen incorrect trials in terms of decodability), we agree with the Reviewer that this particular hypothesis is difficult to assess in the context of the current study and should be further investigated in future studies. We have therefore drawn attention to this fact throughout the manuscript and discuss it in more detail at the end of the Discussion section. However, none of our central claims (i.e., rejection of the alternative hypotheses) rests on the absence of decodability on the unseen correct trials. For example, the rejection of an active conscious maintenance hypothesis requires the absence of response-related activity on all unseen trials (after all, if subjects actively maintained a conscious guess, they would have done so whenever they did not see the target) and, even on the seen trials, there was no evidence for a continuous maintenance of target/response location.

Despite all of our efforts (i.e., inclusion of Bayes’ Factors, converging evidence from univariate and multivariate analyses, ongoing replication of the main null effects), we appreciate that the current study is not without its limitations. The nature of the research question requires long trials and an approximately equal proportion of seen and unseen targets, thereby limiting the number of trials per condition one can obtain within a reasonable time frame. In addition, it necessitates the (statistically impossible) demonstration of null effects. It will therefore be important for future investigations to replicate the current findings, preferably with larger datasets (as, for example, already done in the case of conscious, activity-silent working memory in Wolff et al., 2015, 2017) and/or techniques with better signal-to-noise ratio, such as intracranial recordings. As suggested by the Reviewer, we now included a discussion of this limitation at the very end of the Discussion section and ensured a more careful wording throughout the entire manuscript. What we hope to achieve with the current work is to provide initial results and set a theoretical framework that will hopefully stimulate future research.